# WHERE DO IMAGES COME FROM? ANALYZING CAPTIONS TO GEOGRAPHICALLY PROFILE DATASETS

## ABSTRACT

Building on studies documenting gender and racial biases in vision-language models, recent works show that such models often fail to generate geographically-representative images that accurately reflect different regions around the world. A common concern is that the data used to train these models is not representative, prompting the question: *which parts of the world do these training examples come from?* To answer this question, we develop a system, *GeoProfiler,* which geographically profiles multimodal datasets by mapping image-caption pairs to countries. Using location information from captions, GeoProfiler maps examples to countries with a high precision (0.86). We then apply GeoProfiler to geographically profile the English captions of the LAION dataset for 10 common entities (e.g., house, flag, etc.). We observe the geographical distribution of 8 entities to obey the power law distribution. The United States, the United Kingdom, and India are most represented, appearing in 53.7% of samples. Problematically, African and South American countries are severely under-represented with only 2.0% and 4.3% of images respectively. We also observe a high correlation between a country's GDP and frequency ($\rho = 0.79$). Lastly, we analyze the diversity of images from individual countries, and find that more images does not imply higher diversity.[1]

## 1 INTRODUCTION

Vision-language models (VLMs) (Radford et al., 2021; Li et al., 2023) suffer from gender and racial biases (Hall et al., 2023a; Fraser & Kiritchenko, 2024; Agarwal et al., 2021), making their deployment in real-world applications harmful. Expanding on these findings, recent studies (Basu et al., 2023; Hall et al., 2023b; 2024) highlight that text-to-image models fail to generate images that accurately reflect the surroundings of different geographical regions around the world. These works share concern over the contents of the data used to train such models, raising an important question: *which parts of the world do the training image-text pairs come from*? Answering this question can serve multiple purposes. Data curators can measure and improve the geographical representativeness of datasets, while practitioners can decide among various datasets before training their models. Additionally, auditors can probe relationships between the geographical distribution in the training data and model behavior, similar to previous studies (Razeghi et al., 2022).

Determining image sources can be challenging in practice. Given a multimodal dataset, one may consider predicting which country an image-caption pair belongs to based on the content of the image, its metadata, URL, or the associated caption. However, image-geolocalizers are still imprecise (Vivanco et al., 2023; Astruc et al., 2024) and image metadata often lacks location details; similarly, such information may either be absent or inaccurate in image URLs. Captions on the other hand, describe images and may mention locations. This calls for a system that can map captions to countries, and also determine the fraction of *underspecified* captions (Hutchinson et al., 2022), (i.e., captions without location mentions). Inferring location from a given caption is in itself a difficult task. First, it requires detecting any location mentions within the text, which is highly context dependent (e.g., "Buffalo" can refer to either a place or an animal). The next step is to correctly assign the detected location to its country name, which can be ambiguous (e.g., "Cambridge" is a city in both the US and England).

In this paper, we develop *GeoProfiler,* a system that maps data points in multimodal datasets to their respective countries. We apply this system to analyze common entities that may have diverse

---

[1]We will publicly release our code and other resources to facilitate geo-profiling of datasets.

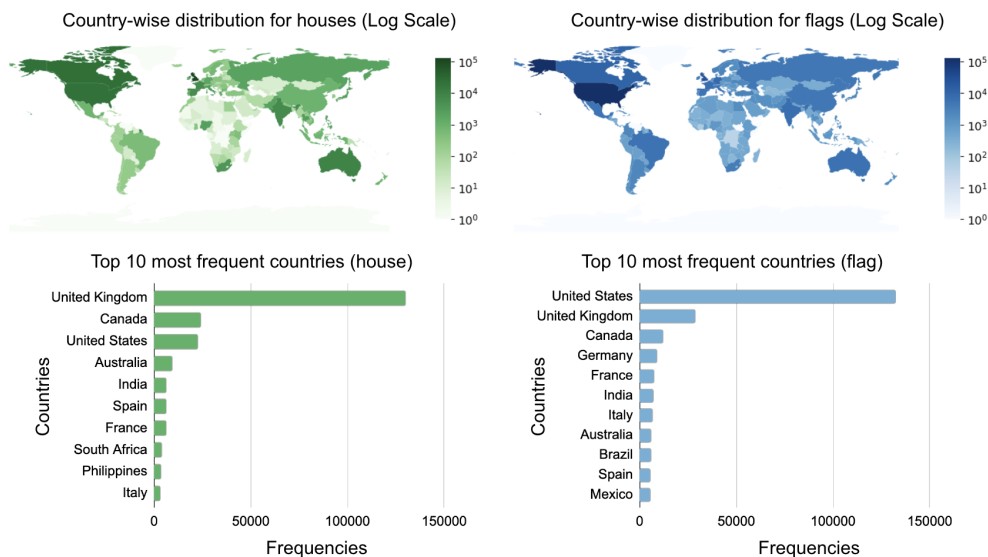

Figure 1: **Distribution of countries for *house* and *flag* in LAION2B-en.** We show the uneven distribution of the different countries around the world for house- and flag-related image-caption pairs in the LAION2B-en dataset, along with the top 10 most frequent countries for these two entities.

appearances globally (e.g., house, road, etc.). While inferring countries, we find that string-matching methods (i.e., detecting locations in a caption from a knowledge base) blindly tag common words as places without accounting for context. In contrast, NER taggers help detect correct locations, but lead to high false negatives. Both methods poorly handle ambiguous places belonging to multiple countries. To address these issues, we use Mixtral-8x7B Instruct (Jiang et al., 2024), a large language model, to map data points to country names (with $0.86$ precision), benefiting from its extensive knowledge and ability to model context.[2] We also observe that images accompanying captions that mention a given entity may not always contain that entity. Hence, we collect a small subset of labeled examples and train an entity-presence classifier to remove irrelevant images (with $0.88$ precision). After filtering for relevant image-caption pairs, *GeoProfiler* (1) analyzes their country-wise geographical distribution, and (2) compares the diversity in images from high and low frequency countries.

Using our tool, we examine LAION2B-en, the subset of the LAION-5B dataset with English captions (Schuhmann et al., 2022), used to train text-to-image models like Stable Diffusion Ramesh et al. (2022). We perform geographical profiling for $10$ entities: *house*, *car*, *flag*, *kitchen*, *road*, *beach*, *hotel*, *bedroom*, *toilet* and *apartment*. *GeoProfiler* uncovers notable differences in the representation of countries (see Figure 1). In fact, the geographical distributions of 8 out of 10 entities obey the power law distribution. We also see a high correlation between frequency of countries and their GDPs ($\rho = 0.79$). South American and African countries are strikingly scarce, only present in $2.0\%$ and $4.3\%$ of the captions respectively. We compare the geographical distributions of the entities with ground truth distributions, and find that for 5 out of 10 entities, more than $50\%$ of countries are underrepresented. Finally, we investigate the diversity in images of a country, and observe that higher frequency does not imply higher diversity. Importantly, note that while geographical biases in model outputs are widely acknowledged, our work conducts a thorough examination of underlying training samples. Broadly, our work facilitates responsible AI development by offering data-centric tools to measure and improve the geographical representations of multimodal datasets.

## 2 RELATED WORK

**Evaluation of Geographical Representation in Models**. Evaluating the geographical representativeness of large-scale models is gaining impetus, particularly in text-to-image (Basu et al., 2023;

---

[2]While we find multiple existing LLMs to be useful in geographically profiling multimodal datasets, we leave the task of holistically benchmarking different LLMs for future work.

Hall et al., 2023b; Naik & Nushi, 2023; Hall et al., 2024) and language generation (Schwöbel et al., 2023; Zhou et al., 2022; Li et al., 2022; Godey et al., 2024), and image search and retrieval (Mandal et al., 2021). A few works also highlight economic and geographical disparities in model performance (Gustafson et al., 2023; De Vries et al., 2019). These studies raise questions about the geographical composition of the corresponding training data, directly motivating our work. We provide further details on these works in Appendix A.1.

**Existing Geoparsing Algorithms.** Extracting location mentions from text and mapping them to countries or geographic coordinates is a well-studied challenge (Middleton et al., 2018; Martínez & Periñán-Pascual, 2020; Hu et al., 2022; Spacy; Kordopatis-Zilos et al., 2017; Luo et al., 2011). Tools like Geoparsepy (Middleton et al., 2018), a multilingual geoparsing system leveraging the OpenStreetMap (OSM)(OpenStreetMap) gazetteer, can be used for recognizing diverse place names, including multilingual or abbreviated ones. Similarly, GazPNE2(Hu et al., 2022) employs deep learning techniques and gazetteers like OSM and GeoNames (GeoNames database) for location extraction. Despite their high precision, these systems often suffer from low recall (Hu et al., 2023), depend heavily on gazetteers, and perform best with formal text, limiting their coverage. Our approach leverages large language models to extract and disambiguate location mentions into country names automatically, eliminating the need for gazetteers and improving coverage.

**Geographical Profiling of Existing Datasets**. Previous studies (De Vries et al., 2019; Shankar et al., 2017; Naggita et al., 2023; Wang et al., 2022; Faisal et al., 2022) assess the geo-diversity of existing datasets, and find that open-source visual datasets like ImageNet (Deng et al., 2009), OpenImages (Krasin et al., 2017) and MS-COCO (Lin et al., 2014) overrepresent North American and European countries (De Vries et al., 2019; Shankar et al., 2017). Furthermore, web-scraped images from African countries have been shown to reflect Western perspectives rather than local perspectives (Naggita et al., 2023). Most of these works obtain geographical annotations from the Flickr API, or utilize external services. Another recent study (Faisal et al., 2022) finds that English speaking and wealthier countries are heavily represented in datasets like SQUAD (Rajpurkar et al., 2016) and MLQA (Lewis et al., 2019). Their work examines geographical representation in language datasets, while ours focuses on vision-language datasets.

## 3 GEO-PROFILING

In this section, we describe the proposed system, GeoProfiler. We first discuss the necessary notation and then provide details about the various components of our system.

**Preliminaries**. Let $\mathcal{D} = \{(x_i, y_i)\}_{i=1}^{N}$ be a vision-language dataset where $y_i$ is the caption accompanying image $x_i$. To analyze the geographical distributions for various entities, we select each entity $n$ to be diverse in appearance, yet well-known and globally relevant (e.g., house).[3] We then randomly sample $\mathcal{D}_n \in \mathcal{D}$ such that for each $(x_i, y_i) \in \mathcal{D}_n$, $n$ is a word in caption $y_i$. GeoProfiler maps each $(x_i, y_i) \in \mathcal{D}_n$ to a tag $c_i \in \mathcal{C}$ (the set of all possible countries including a "no country" tag). For simplicity, we choose countries as our denomination of analysis since finer distinctions like states or cities would significantly increase the complexity of the study. In the following subsections, we describe the the method we use for geo-profiling the captions of $\mathcal{D}_n$ (§3.1). To ensure that we only consider images that contain the entity $n$, we train and use an entity-presence classifier (§3.2).

### 3.1 GEO-PROFILING CAPTIONS

The caption-based geo-profiling involves two steps: (1) detecting locations in captions (e.g., Cambridge), and (2) mapping locations to their countries (e.g. England). We first geographically annotate a dataset of 1000 randomly sampled captions from $\mathcal{D}_n$ (inter-annotator agreement is 90%), and explore several approaches for geo-profiling using this dataset. We observe that string-matching, which matches substrings in captions with locations from a geodatabase (e.g., GeoNames (GeoNames database)), leads to many false positives. Since this approach does not account for context, common entities like "Stock" and "Century" (places in Canada and the US) are tagged as places, resulting in a very low precision of 0.12 and recall of 0.79 on our annotated data. These false positives can be reduced by NER taggers (Spacy) that mark places in a text with a *location* tag (precision is 0.71). However, NER taggers tend to miss location mentions, increasing false negatives (recall is 0.59).

---

[3]We avoid entities that are represented using different words across cultures (e.g., food, clothing).

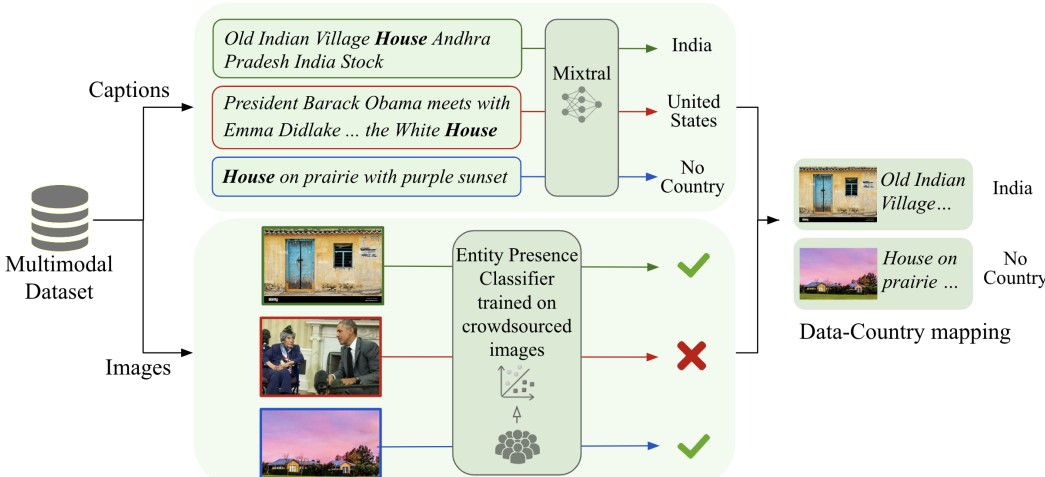

Figure 2: **The proposed GeoProfiler.** Given an image-caption pair of an entity, *GeoProfiler* first maps captions to countries (using Mixtral), then filters out images that do not contain the entity (using the entity presence classifier), and finally returns the countries for resulting image-caption pairs.

Also, the geodatabase lacks information about the likelihood of a place name belonging to a specific country, making it unsuitable for handling ambiguous place names that map to multiple countries.

These challenges in geo-profiling lead us to the realization that the system must be context-aware and possess world-knowledge to accurately extract locations from the text and infer their countries. For this, we explore the use of language models, and find them to be accurate in identifying countries from captions. We compare the performance of several geo-profiling approaches in Table 9 in the Appendix), and find Mixtral-8x7B Instruct (Jiang et al., 2024) to be most effective in mapping the captions to countries (or tagging them as "no country"). This LLM-based method outperforms the string-matching and NER-based algorithms, achieving 86% precision and 82% recall – we use Mixtral-8x7B Instruct for all our experiments in the following sections.[4] Additionally, we also extract 1660 sentences from Wikipedia about marginalized countries (e.g., Tuvalu, Kiribati, etc) and their cities, and find our LLM-based approach to correctly infer the countries 76% of the time (see Appendix A.4). This suggests that our approach could be used to identify a diverse range of countries across the world. We leave a comprehensive comparison of various LLMs that could be applied for this task to future work. Further details on geo-profiling captions are available in the Appendix A.4.

**A Note on Geo-profiling Images**. We also assess the feasibility of inferring countries directly from images. For each entity $n$, we sample 100 images per country, based on silver labels from the Mixtral model (we only consider entity and country pairs whose frequency exceeds 100). We evaluate two state-of-the-art geo-localization models: GeoCLIP (Vivanco et al., 2023) and OSV-5M (Astruc et al., 2024). GeoCLIP achieves an accuracy of 35.5% at the country level and 69% at the continent level. In contrast, the OSV-5M model performs less accurately, with country and continent accuracies of 22% and 56%, respectively. These results highlight the challenges and limitations of existing geo-localization methods in predicting specific countries or even broader continent-level locations from visual data alone. We share the accuracies for individual entities in Appendix A.6. Further research is needed to accurately infer locations from images. Comparatively, LLM-based methods are more effective in inferring locations from captions whenever they contain geographical cues.

## 3.2 ENTITY PRESENCE FILTERING

Recall that the dataset $\mathcal{D}_n$ consists of captions containing the entity $n$ and corresponding images. However, images in $\mathcal{D}_n$ may not always depict the entity $n$. This calls for a filtering step to remove these irrelevant images. We find zero-shot foundation models like CLIP (Radford et al., 2021) to

---

[4]GPT-4o offers slightly higher precision (87%) and recall (91%) for this task. However, given the scale of profiling, we refrain from using GPT-4o due to financial considerations.

be inadequate for this purpose (details in Appendix A.3.4). Therefore, we use a entity presence classifier, a binary classifier that predicts whether a given entity $n$ is present in an image.

To train this classifier, we sample a set of approximately 550-650 images from $\mathcal{D}_n$. To ensure these images are geographically diverse, we use the predictions obtained from geo-profiling the captions to sample images from various regions (Economic Regions) and economic stratas (income, None), categorizing countries into 4 income groups and 17 geographical regions (details in the Appendix A.3). We select 15 images from every possible income-region combination, creating a geographically and economically diverse set. We then recruit crowd workers from Prolific (prolific, None) to mark images in which the given entity is visible. Each image is shown to 3 annotators, and the final annotation is determined by majority voting. For a 15 minute survey, annotators are paid \$2.45. The inter-annotator agreements are presented in Appendix A.3.3. Of the annotated images, 100 form the test set, and the rest are used for training. We train an SVM classifier with CLIP image features and image annotations as labels, which achieves a high precision of 0.88. Finally, using this model, we filter out irrelevant image-caption pairs from $\mathcal{D}_n$. An overview of our geo-profiling approach is depicted in Figure 2.

## 4 A Case Study: the LAION2B-en Dataset

Geographical profiling can offer answers to several important questions about the contents of datasets, and models trained using them. As a case study, we consider the LAION2B-en dataset, and ask:

**RQ1**: What is the geographical distribution of globally-relevant entities? Are some of the countries over (or under) represented in the data?

**RQ2**: For an entity $n$, are the data points in $\mathcal{D}_n$ equally diverse for different countries? Furthermore, does higher frequency ensure greater diversity?

We begin by describing the chosen dataset, LAION2B-en, and introducing the entities we select. We then answer the above questions in subsections 4.1, 4.2, and 4.3. Qualitative examples of the images for different entities and countries can be seen in Appendix A.7.

**Dataset**. We work with the LAION2B-en dataset (i.e., the subset of the dataset with English captions), used to train models including Stable Diffusion and Midjourney (Rombach et al., 2022; Midjourney). It contains 2.3 billion image-caption pairs collected from the Common Crawl database (Common Crawl). For entity $n$, we randomly sample $\mathcal{D}_n$ of size 1M, ensuring that each caption in $\mathcal{D}_n$ contains $n$, using the WIMBD tool (Elazar et al., 2024). We select 10 entities: *house, flag, car, beach, kitchen, road, hotel, bedroom, toilet*, and *apartment*. The selected entities are those that one might interact with in their daily lives, but their usage and appearances may vary across regions.

| % | House | Flag | Car | Kitchen | Beach | Rd | Hotel | Bedroom | Toilet | Apt | Overall |
|---|---|---|---|---|---|---|---|---|---|---|---|
| Underspecified | 41.3 | 30.6 | 85.6 | 87.7 | 45.6 | 59.4 | 26.5 | 82.7 | 88.9 | 20.6 | 46.1 |
| Top 10 | 49.4 | 40.5 | 11.9 | 10.4 | 36.4 | 29.3 | 45.0 | 12.9 | 9.0 | 43.8 | 38.8 |
| Remaining | 9.3 | 28.9 | 3.5 | 1.9 | 18.0 | 11.3 | 28.5 | 4.4 | 2.1 | 35.6 | 5.1 |

Table 1: **Geographical distribution for each entity**. We present the percentage of data points that a) have *underspecified* captions, b) belong to the *top* 10 most frequently occurring countries, and c) belong to the *remaining* countries. Overall, we observe that $46.1\%$ captions are underspecified, $38.8\%$ are from the top 10 most frequent countries, and only $5.1\%$ are from the remaining countries.

### 4.1 Distribution-Based Analysis

Based on the predictions from GeoProfiler, we study the geographical distribution of $\mathcal{D}_n$ for each entity $n$ and present our analysis below. For each entity, we show the percentage of a) underspecified image-caption pairs, b) those belonging to the top 10 most frequent countries, and c) those assigned to the remaining ones, in Table 1.

**Underspecified Image Captions**. Overall, $46.1\%$ captions are tagged as underspecified by GeoProfiler across entities. Four entities, including kitchen and toilet exhibit strikingly high percentages of underspecification (over $80\%$). This is unsurprising as people are unlikely to include locations in their

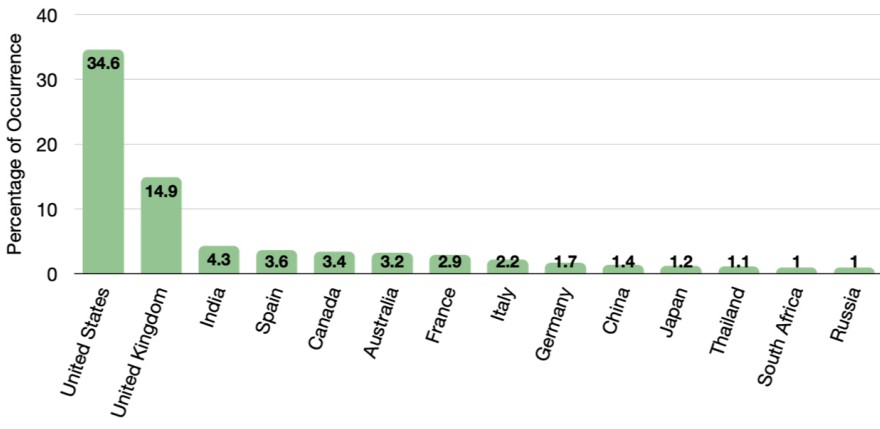

Figure 3: **Percentage of occurrence for the most frequent countries (averaged across entities).** We observe that the overall distribution of countries follows a power law distribution (p-value $< 0.05$).

descriptions of toilets and kitchens. In contrast, apartment, hotel, flag have the lowest percentages ($< 31\%$), indicating that their captions frequently contain location indicators. The high percentages of underspecified captions highlights a fundamental challenge in geo-profiling a dataset.

**Location-specified Captions**. To answer **RQ1** posed in §4, we find that the country-wise distribution is highly skewed in captions. Across entities, we find countries such as the US, UK, India, Spain, Canada, etc. to be among the top 10 most frequent countries (see Fig. 3). In fact, the percentage belonging to the top 10 most frequent countries is at least twice that of all other countries combined for the majority of entities (the exceptions being apartment, flag, and hotel), as shown in Table 1. Further, the geographical distributions for all entities individually (except hotel and beach) follow the power law distribution (p-value $< 0.05$). We see the same pattern for the overall distribution obtained from combining the data from all entities. On a continent level, Europe and North America have the highest representation across entities (37.6% and 33.5% of all samples respectively), followed by Asia (18.5%). This observation aligns with those from past studies (De Vries et al., 2019; Shankar et al., 2017) which find overrepresentation of North American and European countries in other datasets as well. In contrast, Oceania, Africa, and South America only represent 10.4% of the image-captions (more details in Appendix A.5).

**Correlation with GDP (nominal).** We find a strong positive correlation ($\rho = 0.79$) between the frequency of countries summed across all entities and their nominal GDP (Gdp reference data), as shown in Figure 4. Individually, 5 out of 10 entities: car, flag, hotel, apartment, and toilet have strong positive correlation ($\rho \geq 0.8$). House is the only entity for which the association is weak ($\rho = 0.24$). We also find a weak positive correlation between the population of the countries and the number of examples in the dataset ($\rho = 0.24$). Overall, our findings suggest that wealthier countries exhibit higher frequency across entities.

### 4.2 GEOGRAPHICAL REPRESENTATIVENESS

To examine the over and underrepresentation of countries, as motivated in **RQ1**, we need to measure representation with respect to some reference. For instance, an entity like beach should be associated with countries with coastlines, rather than landlocked ones, i.e., the geographical distribution should reflect the true occurrence of the entity worldwide. Hence, we define geographical representativeness of a country to be the extent to which its distribution in $\mathcal{D}_n$ differs from a ground truth reference distribution. We gather such distributions from credible sources (e.g., Wikipedia and United Nations, details in Appendix A.9). Let $p(c|x, y, n)$ represent the distribution of countries given the image-caption pair $(x, y)$ in $\mathcal{D}_n$ and $p_{\text{true}}(c|n)$ be the ground truth distribution of countries for the entity $n$. We define the geographical representativeness (GR) of a country $c$ with respect to entity $n$ as:

$$\text{GR}(c, n) = \frac{p(c|x, y, n)}{p_{\text{true}}(c|n)}$$

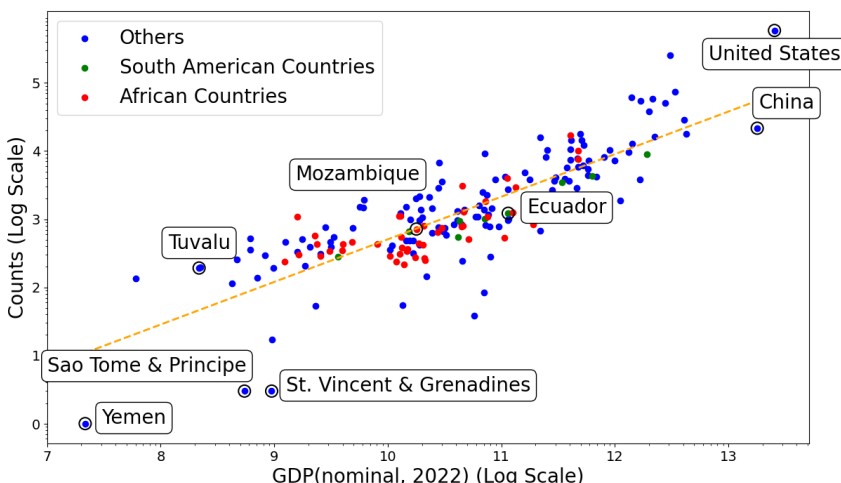

Figure 4: **Correlations of counts across all entities and GDP (nominal)**. We find a high correlation ($\rho = 0.79$), which shows that wealthier countries are more represented across entities.

| | House | Flag | Car | Kitchen | Beach | Road | Hotel | Bedroom | Toilet | Apartment | Avg |
|---|---|---|---|---|---|---|---|---|---|---|---|
| Under | **56.5** | **68.8** | 26.8 | **58.7** | 34.0 | 23.4 | 41.8 | 46.6 | **55.5** | **55.0** | 46.7 |
| Over | 8.0 | 2.6 | 13.4 | 11.9 | 28.0 | 19.0 | 14.4 | 22.4 | 13.9 | 8.4 | 14.2 |

Table 2: **Geographical Representation for each entity**. We show the percentage of countries that are underrepresented (Under) and overrepresentated (Over) for each entity. On average, $46.7\%$ countries are underrepresented, whereas $14.2\%$ are overrepresented across entities.

We consider $c$ to be *overrepresented* if GR$(c, n) > $ r, and *underrepresented* if GR$(c, n) < \frac{1}{r}$, where r $\geq 1$ is a hyperparameter. Intuitively, a country is called overrepresented if it is at least r times more likely to occur in $\mathcal{D}_n$ as compared to the real world. Following previous works on geographical erasure (Schwöbel et al., 2023), we set r $= 3$. We show the effect of other values of r on our analysis in Appendix A.9.

**Findings**: We observe that for 5 entities, more than $50\%$ of the countries are underrepresented, as shown in Table 2. On average, $46.7\%$ countries are underrepresented, and $14.2\%$ are overrepresented. Across most entities, UK, Australia, and Singapore are consistently overrepresented, while Kazakhstan, Uzbekistan, and Chile are consistently underrepresented. We believe that such information can help curators create geographically balanced datasets.

### 4.3 DIVERSITY ANALYSIS

We measure the diversity of images and captions of different countries, and examine the relationship between diversity and frequency (**RQ2**). Given any set $S = \{x_1, x_2, \cdots x_N\}$ of size $N$, as well as a feature encoder $f$, we define the diversity of this set to be the mean squared root distance between the features of each sample and the mean feature vector (following previous works (Fan et al., 2023; Boutin et al., 2023)). Formally,

$$div(S) = \sqrt{\frac{1}{N} \sum_{i=0}^{N} dist(f(x_i), \bar{f})^2}$$

where $f(x_i)$ is the L2-normalized feature vector (Liu et al., 2024) of $x_i$ obtained from encoder $f$, $\bar{f} = \frac{1}{N} \sum_{i=0}^{N} f(x_i)$ is the mean vector calculated over all data points in $S$, and $dist(f(x_i), \bar{f})$ is the Euclidean distance between the vectors. Higher $div(S)$ means that the feature vectors are more spread out, indicating higher diversity. In practice, we compute diversity for data points belonging

|       | House | Flag | Car | Kitchen | Beach | Road | Hotel | Bedroom | Toilet | Apartment |
|-------|-------|------|-----|---------|-------|------|-------|---------|--------|-----------|
| $\rho_{\text{fi}}$ | -0.5 | 0.18 | 0.1 | -0.16 | 0.33 | 0.07 | 0.19 | -0.03 | -0.21 | 0.11 |

Table 3: **Frequency and diversity correlations.** For each entity, we compute the correlation between the frequency of countries and diversity scores of the images ($\rho_{fi}$). We notice that the majority of entities exhibit very little correlation.

to a specific country and entity, but only consider countries that have more than $100$ images for a given entity. For each entity $n$, we define its overall diversity as the average diversity across countries. To calculate the diversity of the images in the dataset, we use DINOv2 (Oquab et al., 2023) as the feature encoder $f$, which has been shown to align with human perceptions (Hall et al., 2024).

**Entity-wise Diversity of Images.** Here, we calculate the overall diversity for the images belonging to each of the studied entities. We find that flag, house, car, road, hotel, and apartment exhibit similar diversity scores, with flags having the highest score ($0.83$). Kitchen and bedroom are the least diverse with values $0.61$ and $0.62$ respectively. We present the diversity scores for each entity in Table 16 (Appendix); the average diversity score across all entities is $0.75$. Note that all findings are statistically significant (p-value $< 0.01$).

**Diversity vs. Frequency**. We investigate if images from frequently occurring countries tend to be more diverse than those from countries with low frequency. We define $\rho_{fi}$ as the Pearson's correlation coefficient of country-wise frequency values with the images. However, we find no clear relation between diversity and frequency across entities with average $\rho_{fi} = 0.008$. We share examples of images from high and low diversity countries in Appendix Figures 9 and 10. The detailed results on correlation between diversity and frequency of the images are presented in Table 3.

Additionally, for each entity $n$, we also compute (1) caption diversity, to study its relationship with frequency, and (2) the diversity of different countries generated by Stable Diffusionv1.3 (Ramesh et al., 2022) (trained on LAION2B-en), to study the relationship between training and generated image diversity (see Appendix A.10 for more details).[5]

### 4.4 FREQUENCY VS HUMAN RATINGS

A recent study inspects the geographical representativeness of Stable Diffusion, where humans rate how well generated images reflect the surroundings in their respective countries (Basu et al., 2023). This study is conducted across 27 countries with 10 commonly-occurring entities, 5 of which overlap with our chosen ones. The study also discusses how representativeness differs for generations with underspecified prompts (e.g., `High definition image of a {entity}`) and country-specific prompts (e.g., `High definition image of a {entity} in {country}`). We follow up on this work and study the relationship between frequencies of countries in the training dataset and human ratings of geographical representativeness of generated images. For generations with unspecified prompts, we observe a positive correlation with frequency for flag, kitchen, and beach ($\rho > 0.4$), indicating that human ratings of geographical representativeness for different countries are proportional to the frequency of occurrence in the training data (see Table 4). However, the correlation is weak for house and road. A similar pattern is seen for the country-specific generations, although the correlation reduces for all entities except house. We feel this happens as the explicit mention of the country name in the prompts increases the scores for each country, not necessarily proportional to their frequencies.

### 5 LIMITATIONS

In this section, we discuss important limitations of our work. As a case study, we apply GeoProfiler on image-text pairs of the LAION2B-en (Schuhmann et al., 2022) dataset (used to train popular text-to-image models), and demonstrate the geographical distributions of images with *English* captions. The LAION-5B dataset includes a multilingual subset of 2.2B image-caption pairs spanning 100 languages (Schuhmann et al., 2022). We believe that analyzing this dataset may help uncover

---

[5]We recognize that feature encoders may exhibit their own biases, and investigate this in Appendix A.10.

| Beach | | House | | Road | | Flag | | Kitchen | |
|---|---|---|---|---|---|---|---|---|---|
| Under | Spec | Under | Spec | Under | Spec | Under | Spec | Under | Spec |
| 0.48 | 0.28 | 0.1 | 0.26 | 0.15 | -0.06 | 0.9 | 0.58 | 0.48 | 0.33 |

Table 4: **Frequency vs Human Ratings**. Pearson's Correlation Coefficient between the frequency of countries in the training data and human ratings of Stable Diffusion generations (Basu et al., 2023), for both underspecified (Under) and country-specific (Spec) prompts, on 5 entities and 27 countries. We observe clear correlations for 3 nouns in the underspecified scenario, while for most nouns, the correlation reduces for the country-specific prompts.

image-text pairs from non-English-speaking regions that are underrepresented in the English subset. However, a preliminary analysis indicates that geoprofiling such captions is challenging due to the performance disparities between multilingual and English-based LLMs (see Appendix A.8), and requires further research to address this gap. Moreover, datasets used to train other prominent models like DALL·E 2 (Ramesh et al., 2022) and Imagen (Saharia et al., 2022) are closed-source. Although we can reasonably expect some overlap in data curation practices and training distributions, the extent to which our findings hold broadly across datasets is unclear. To this end, we will open-source our code to assist in performing evaluations for other datasets.

While our study focuses on 10 entities, a future direction is to extend it to those that are country-specific, for a more nuanced picture of geographical representation (e.g., food is often discussed with dish names like risotto, biryani, paella, etc. rather than the word "food"). Also, the language models we use for geo-profiling captions are likely unaware of many places around the world. As discussed in subsection 3.1, geo-profiling is challenging, and we urge future research to quantify the geographical biases encoded by models. A similar analysis is required to quantify the biases in other pretrained models we use to calculate the diversity values (e.g., DiNO v2). Likewise, the crowdworkers employed to verify the presence of specific entities in images may not be familiar with global variations in how entities are depicted.

While LAION2B-en (Schuhmann et al., 2022) (with a CC-BY 4.0 license) is no longer publicly available, we began this project when it was still available for download. We believe that it is important to geo-profile this dataset, as models trained on it are still widely used. We are confident that the chosen subset of LAION is innocuous and excludes the problematic content that led to its removal. Additionally, it is important to highlight that the GeoProfiler exclusively outputs geographical predictions for captions with geographical cues. Estimating locations from underspecified image-caption pairs remains a significant challenge. Firstly, only $0.71\%$ of the analysed images of the LAION2B-en dataset contain GPS metadata. Moreover, our preliminary analysis also reveals the limitations of current image geolocalization methods in accurately extracting location information. To address this, we suggest that future data curators document the origins of images meticulously (whenever possible), to enable supervised learning based approaches for geo-localization. Lastly, it is to be noted that while there are privacy considerations that arise with geo-profiling, we perform a coarse-grained investigation and do not analyze people or specific GPS coordinates.

## 6 CONCLUSION

In this work, we proposed *GeoProfiler,* a system that maps image-caption pairs of vision-language datasets to corresponding countries. We investigated the geographical distribution of the LAION2B-en dataset (the subset of the LAION-5B dataset with English captions) with respect to 10 entities, and found that $46.1\%$ of the total captions were *underspecified*. For the remaining samples, we observed that the geographical distributions for 8 out of 10 entities obey the power law distribution. When comparing the geographical distributions of each entity with ground truth reference distributions, we found that for 5 entities, more than $50\%$ of the countries exhibit underrepresentation. Across all entities, the US, UK, and India are the most frequent countries, while many countries in South America, Oceania, and Africa appear infrequently. We also explored the country-wise diversity of images for each entity, and discovered that the frequency of a country does not correlate with its diversity. We hope that GeoProfiler encourages and enables greater transparency into the geographical representation of large-scale multimodal datasets.

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

## A  APPENDIX

In this appendix, we begin by discussing a few related works (subsection A.1) and the compute resources we use for our tool (subsection A.2). We next discuss the finer details about the entity-presence filtering step of the proposed system, including the details on the dataset selection for the crowdsourcing task, the survey details with inter-annotator agreements, and comparison with other methods (subsection A.3). We next discuss the different approaches we explored for geo-profiling the captions, and their comparisons with our approach (subsection A.4). Thereafter, we present the distribution of continents in the subsets of the LAION2B-en dataset (Schuhmann et al., 2022) that we choose for each entity (subsection A.5), followed by further analysis on geo-profiling the images directly A.6, some qualitative examples (subsection A.7), and a discussion on the ability of existing LLMs to geo-profile multilingual captions (subsection A.8). Finally, we end our discussion with some additional details and results on the geographical representativeness of countries (subsection A.9) and the diversity in their images (subsection A.10).

### A.1  EXTENDED RELATED WORK

In this subsection, we expand on some of the prominent works that we discuss in the main paper (Section 2). Several works point out the disparity in model generations based on different geographies. Hall et al. (Hall et al., 2023b) find that popular text-to-image generative models exhibit lower diversity and realism when generating images from African and Western Asian countries as compared to European images. Moreover, another work (Hall et al., 2024) discovers that the perception of geographical representation varies region to region over the globe, making evaluation of such models more difficult. Some works also highlight how countries having large English-speaking populations are underpredicted by language models (Schwöbel et al., 2023). Not only do these biases affect generative models, but also the state-of-the-art object recognition models. For example, Gustafon et al. (Gustafson et al., 2023) show that performances of popular models like CLIP (Radford et al., 2021) degrade with decreasing income level. Similar performance drops are seen in other works as well (Ramaswamy et al., 2024). These papers point out the disparities of models trained on different modalities with respect to various parts of the world – thus, highlighting a big drawback of current machine learning models. Analysing the geographical composition of the datasets used to train these models is crucial to assess their behavior. As discussed in Section 2 (main paper), there are several works (De Vries et al., 2019; Shankar et al., 2017; Faisal et al., 2022) on exploring the geographical distributions of popular image and text datasets that uncover the overrepresentation of North American and European countries in existing image and text datasets. REVISE (Wang et al., 2022) measures biases in image datasets with respect to objects, people and also geographies, however it relies on external availability of geographical data. Our proposed tool focuses on multimodal datasets, and maps countries from location mentions in the relevant image-captions. We are hopeful that the analysis we perform in this paper will guide future dataset curators and model auditors to evaluate other datasets as well as understand model predictions and generations.

**Existing Geo-diverse Datasets**. Existing datasets overrepresent Western, English-speaking countries (De Vries et al., 2019; Shankar et al., 2017). Thus, several geo-diverse datasets have been recently proposed. For example, *DollarStreet* (Rojas et al., 2022) and *GeoDE* (Ramaswamy et al., 2024) are geo-diverse image datasets collected through manual and crowdsourcing efforts. Other multi-cultural geo-diverse image and language datasets include GeoYFCC (Dubey et al., 2021), MaRVL (Liu et al., 2021), GD-VCR (Yin et al., 2021), CultureAtlas (Fung et al., 2024), *GeoNet* (Kalluri et al., 2023), among others.

## A.2 Resources used by GeoProfiler

We use a NVIDIA RTX A6000 GPU card to run the Mixtral model and other benchmarking tasks like using the BLIPv2 VQA model (Li et al., 2023) and the CLIP model (Radford et al., 2021) to evaluate on the crowd-annotated dataset. No GPU is required for any of the other components of GeoProfiler.

## A.3 Entity-Presence Filtering - Further Details

The entity-presence filtering removes images that are irrelevant to the entity in question. Here, we present the different steps employed in this process: a) Creating a small dataset for annotation, b) Conducting a survey on crowdworkers with the annotated images. We next provide the overall inter-annotator agreements for each entity.

### A.3.1 Annotation Dataset Creation

We first select a small subset of images for each entity, and then hire crowdworkers to annotate them. To ensure that the selected images are geodiverse, we divide the globe into 17 regions as described by United Nations and 4 income groups as described by World Bank (see subsection 3.2). The 17 regions are as follows: *['Northern Africa', 'South America', 'Sub-Saharan Africa', 'Western Europe', 'Australia and New Zealand', 'Southern Europe', 'Western Asia', 'Eastern Europe', 'Caribbean', 'Central America', 'Eastern Asia', 'Oceania (excluding Australia and New Zealand)', 'Northern America', 'Southern Asia', 'Central Asia', 'South-eastern Asia']* whereas the Income Groups are *['Low Income', 'Lower Middle Income', 'Upper Middle Income', 'High Income']*. Using the predictions from the geo-profiling stage of GeoProfiler, we select equal number of images from each region and income group (or all images in case the number of images is less for a certain region and income group combination). For proper evaluation, our test set has both in-distribution (ID) data (i.e., images from countries already seen in the training set), and out-of-distribution (OOD) data (i.e., images from countries not seen in the training set). The following set of countries make up the OOD test set: *['China', 'Chile', 'New Zealand', 'Philippines', 'Spain'].*

### A.3.2 Survey Details

For each entity $n$, we host a survey with the selected images, and appoint 3 annotators from the Prolific platform (prolific, None) to mark the images containing the entity. To each annotator, we ask the following question: "In this survey, you will be provided with a survey website. Click on the Generate Survey button. When you click on the button, a set of 25 images will show up. Some of them may not show up, we suggest you ignore those images. These images are possible {n} images. For the valid images, select the images where a {n} is visible in the image. After you have made the selection, click on the Update Annotations button which will submit your current results and show you the next set of images. Please wait some time after clicking the Update Annotations button for old images to be replaced with new set of images. At the end, you will get a survey code, please provide us the survey code for our reference." We provide the screenshots of the instructions and the images used in the survey for houses in Figures 5 & 6.

### A.3.3 Inter-Annotator Agreements

For each entity $n$, we show the crowdworkers $550 - 650$ images. To ensure that the annotators are trustworthy, we initially conduct a survey on a smaller dataset (which we annotate by ourselves), and verify the honesty of the crowdworkers by matching their annotations with ours. Finally, we filter out the annotators which have an absolute agreement $< 70\%$. For the real survey, we employ 3 annotators for each entity, and select the final annotation by majority voting among the received votes. Each such survey is conducted for a duration of around 15 minutes, and each crowdworker is paid at the rate of \$9.85 per hour. Here, we discuss the inter-annotator agreement among the crowdworkers for each entity, using Fleiss' Kappa (Fleiss, 1971). Fleiss' Kappa $\kappa$ measures the reliability of agreement

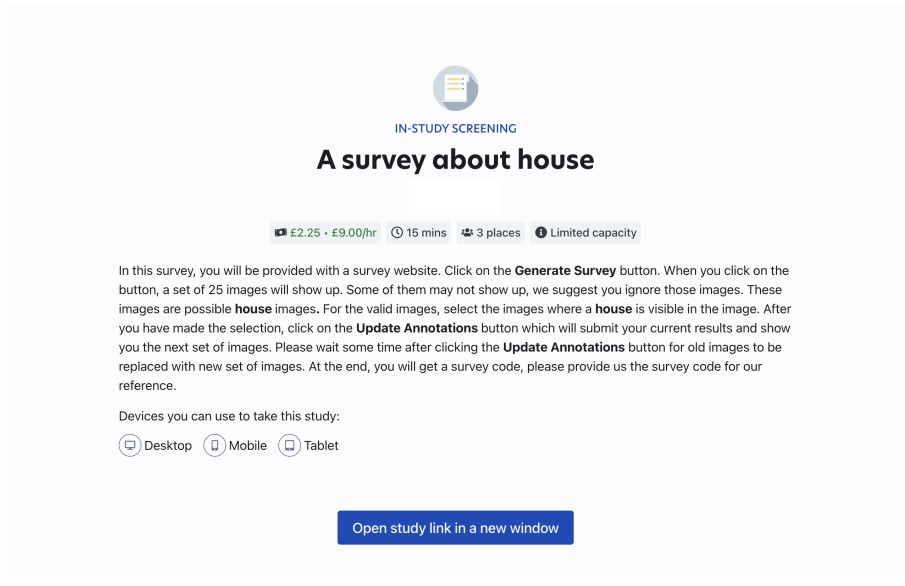

Figure 5: **Survey Instructions**. Given an entity $n$, we show an instruction sheet for the same, where we ask annotators to mark the shown images if they contain the same entity.

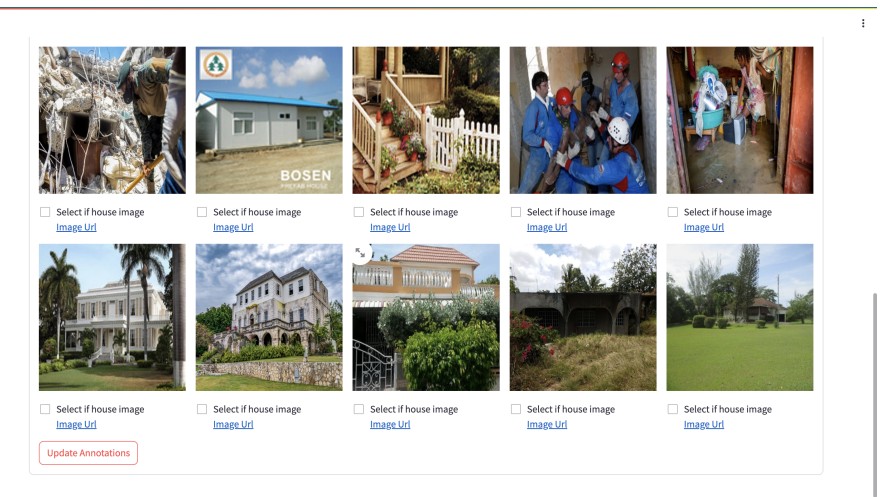

Figure 6: **Survey Main Page.** We show a set of 25 images per page and prompt the user to press the Update Annotations button after the selections to show next set of images

between multiple raters for categorical data and is ideal for our use case as it handles fixed numbers of raters with complete ratings effectively. According to Fleiss's interpretation, $\kappa$ between $0.41 - 0.60$ is considered moderate, a $\kappa$ between $0.61 - 0.80$ as substantial and a $\kappa$ between $0.81 - 1$ as almost perfect agreement. We further measure average agreement between every pair of annotators for both positive and negative classes. To calculate the agreement values for a class, we find the percentage of images for which the pair of annotators agree. Specifically, we define the agreement between a pair of annotators $(i, j)$ as $(\frac{A^c_{(i,j)}}{N^c_j}) * 100$ where $A^c_{(i,j)}$ is the number of images where both annotators $i$ and $j$ agree on the class $c$ and $N^c_j$ refers to the total images where annotator $j$ marked it as class $c$. We find the average over all combinations of ordered pairs of annotators for the class $c$ for each entity $n$ and report this value. We also report the overall agreement which is calculated as the average of the class-wise values. We present the $\kappa$ values, average agreement for classes $0$ and $1$ and also the overall agreement value, averaged across these classes for each entity in Table 5. Further, for each

| entity | $\kappa$ | Avg agreement for class 0 (%) | Avg agreement for class 1 (%) | Overall Agreement (%) |
|---|---|---|---|---|
| house | 0.7 | 85.2 | 86 | 85.6 |
| car | 0.8 | 87.1 | 90.2 | 88.6 |
| road | 0.7 | 84.6 | 81.6 | 83.1 |
| beach | 0.6 | 86.5 | 79.8 | 83.2 |
| flag | 0.6 | 68.1 | 91.3 | 79.7 |
| hotel | 0.6 | 84 | 74.8 | 79.4 |
| toilet | 0.8 | 93.8 | 85.7 | 89.8 |
| bedroom | 0.8 | 97.2 | 85.8 | 91.5 |
| kitchen | 0.8 | 92.5 | 84 | 88.2 |
| apartment | 0.5 | 62.9 | 93.4 | 78.2 |

Table 5: **Fleiss' Kappa and agreement scores across annotators**. After conducting the surveys, we find the mean agreement (over both classes) among all pairs of crowdworkers to be $84.7\%$.

| | Apart-ment | Kit-chen | Ho-use | Ho-tel | Toi-let | Bed-room | Ro-ad | Be-ach | Car | Fl-ag |
|---|---|---|---|---|---|---|---|---|---|---|
| Total Num | 577 | 526 | 651 | 660 | 554 | 585 | 640 | 590 | 634 | 660 |
| Present | 507 | 172 | 334 | 293 | 161 | 152 | 300 | 244 | 370 | 535 |
| Absent | 70 | 354 | 317 | 367 | 393 | 433 | 340 | 346 | 264 | 125 |

Table 6: **Entity-wise details on the number of images annotated**. Alongside the number of images annotated, we also report the number of images with and without the entity.

entity, the total number of images annotated and the number of images with and without the entity is presented in Table 6. It is to be remembered that to curate the annotation dataset for each entity, we uniformly sample geo-diverse images for a given entity from the country-tagged image-caption pairs from a combination of 17 regions and 4 income groups across the world. Since an equal number of images from all region-income group combinations may be unavailable, the total number of images varies across entities.

### A.3.4 BENCHMARKING THE ENTITY-PRESENCE CLASSIFIER

In section 3.2, we mention that we train an SVM model using the CLIP features of the crowd-annotated images to predict if the entity is present or absent in them. Here, we compare its performance with other methods: a) the zero-shot CLIP (Radford et al., 2021) model, b) the zero-shot BLIPv2 (Li et al., 2023) model, c) an SVM model trained on the geo-diverse *GeoDE* (Ramaswamy et al., 2024) dataset.

The $f1$-score for each model is presented in Table 7. Recall that the test set consists of both ID and OOD subsets. For the CLIP model, we compare each image of a certain entity by the following text prompts: "Photo of a {entity}", and "Not a photo of a {entity}", and based on the similarity of the image with these two prompts, we assign label 0 to it if it is more similar to the latter prompt, else we assign 1. We also evaluate the VQA model of BLIP, and for each image of an entity, we ask the following question: "Is this a photo of any {entity}". We again assign 0 to the image if the answer is 'no', otherwise 1 is assigned. Additionally, we evaluate an SVM model trained on the GeoDE dataset, which has images from different parts of the world with respect to multiple entities. The only entities that are common with our paper, are house, flag and car. For all the compared methods involving CLIP, BLIP and the GeoDE dataset respectively, we notice that the classifier trained on the crowd-annotated dataset surpasses them in terms of $f1$-score for both the ID and OOD subsets. This demonstrates the necessity of the crowd-annotation step in order to train the entity presence classifier.

| entity | CLIP | | BLIP | | SVM (GeoDE) | | SVM (Ours) | |
|---|---|---|---|---|---|---|---|---|
| | ID | OOD | ID | OOD | ID | OOD | ID | OOD |
| House | 0.75 | 0.78 | 0.79 | 0.89 | 0.77 | 0.88 | **0.85** | **0.92** |
| Flag | 0.66 | 0.72 | 0.84 | 0.82 | **0.95** | 0.94 | 0.92 | **0.94** |
| Car | 0.82 | 0.76 | 0.88 | 0.74 | 0.83 | 0.88 | **0.89** | **0.98** |
| Kitchen | 0.70 | 0.76 | 0.78 | 0.80 | NA | NA | **0.93** | **0.88** |
| Beach | 0.80 | 0.61 | 0.80 | 0.67 | NA | NA | **0.85** | **0.81** |
| Road | 0.63 | 0.59 | 0.75 | **0.91** | NA | NA | **0.80** | 0.88 |
| Hotel | 0.75 | 0.76 | 0.85 | 0.78 | NA | NA | **0.89** | **0.90** |
| Bedroom | 0.37 | 0.62 | 0.67 | 0.80 | NA | NA | **0.82** | **0.90** |
| Toilet | 0.65 | 0.67 | 0.73 | 0.71 | NA | NA | **0.93** | **0.75** |
| Apartment | 0.91 | 0.87 | 0.95 | 0.88 | NA | NA | **0.95** | **0.98** |

Table 7: **Performance of image recognition models across entities**. We evaluate the *f1-scores* of positives on crowd-annotated ID and OOD test sets for each entity on: a) CLIP (Radford et al., 2021) zero-shot prompting with a negative and positive prompt, b) the BLIPv2 (Li et al., 2023) model, c) SVM model trained on the GeoDE dataset (Ramaswamy et al., 2024), d) SVM model trained on the crowd-annotated training set. For all 10 entities, the SVM classifier trained on the crowd-annotated dataset outperforms the other three methods.

### A.3.5 EXAMPLES OF IRRELEVANT IMAGES

We show examples of relevant and irrelevant images as identified by our entity-presence classifier for each entity in Table 8, further demonstrating the requirement of the entity-presence classifier in the proposed tool.

Table 8: **Irrelevant & Relevant Images**

| | Irrelevant | Relevant |
|---|---|---|
| House |  |  |
| Flag |  |  |

Continued on next page

Table 8 – continued from previous page

| | Irrelevant | Relevant |
|---|---|---|
| Car |  |  |
| Kitchen |  |  |
| Road |  |  |
| Beach |  |  |
| Hotel |  |  |

Continued on next page

Table 8 – continued from previous page

| | Irrelevant | Relevant |
| --- | --- | --- |
| Toilet |  |  |
| Bedroom |  |  |
| Apartment |  |  |

### A.4 FURTHER DETAILS ON GEO-PROFILING THE CAPTIONS

**Comparison of different geo-profiling methods**. Subsection 3.1 in the main paper discusses the challenges faced by the GeoProfiler in predicting countries for image-text pairs. We explore a number of alternatives to geo-profile the captions, which we describe below. For the same, we randomly sample 1000 captions from the LAION2B-en dataset and geographically annotate them ourselves – i.e., by looking at each caption, we manually predict the country name possibly associated with the caption. Based on the annotations of 5 English-speaking participants, the average inter-annotator agreement is found to be 90%. Using this dataset, we evaluate several methods, which we describe below. We also analyse the effectiveness of other existing LLMs in the task of geoprofiling: llama3.1 8B instruct [6], gemma-2-9b-it [7] and gpt-4o [8].

- **String Matching**: It searches for substrings in the captions that can be potential places as listed in the geodatabase we describe in subsection 3.1 in the main paper. While it is a fast algorithm, it can lead to a lot of false positives on account of ignoring the context.

- **NER taggers**: Instead of searching for substrings in the captions blindly, we use NER taggers to detect locations. Specifically, we choose the 'GPE' tags returned by the spacy NER taggers (Spacy) as place names. Spacy provides four models: small, medium, large and transformers. To take full advantage of all models, we pass the caption through each of them iteratively, until a place name is captured by one of them. Finally, the country is identified by searching for the country associated with the detected location with the help of

---

[6] https://huggingface.co/meta-llama/Llama-3.1-8B-Instruct
[7] https://huggingface.co/google/gemma-2-9b-it
[8] https://openai.com/index/hello-gpt-4o

the geodatabase. This method is more precise than string matching methods, as it can more accurately identify locations. But its recall is less, as very often NER taggers (including all their models) tend to miss place mentions in a given text.

- **Mixtral LM**: This is the final method we use to identify country names from captions. The method is described in subsection 3.1. To prompt the model, we use the following instruction for a caption $y$: "`Given the caption: {y} , Identify the country associated with the location mentioned in the caption. If no location or country is specified, output 'No'. Use the format 'Output: (country name/No)'. The output should only be 'No' or the name of the country.`". We prefer this language model as it is found to follow the given instruction, and returns the exact country name (or 'no') for majority of the captions.

- **Effectiveness of other LLMs**: For the Gemma model, we use the same prompt as that for the Mixtral LM. However, we find its performance to be relatively worse than the Mixtral model. For the llama and the gpt4-o models, we set the assistant/system prompt as "`You are a geotagging agent who tags each given text to a country. The only output you give is either the coutry name or 'NO' in case the text cannot be tagged to a country`". For the user prompt, we send the actual caption. While the llama model underperforms slightly as compared to the mixtral model, gpt4-o outperforms the same. We finally choose the Mixtral model for the GeoProfiler as it is both open-source and effective.

- **NER taggers + Mixtral LM**: In case of resource constraints, one can use a hybrid system by invoking the Mixtral model for captions where the NER tagger is unable to detect any location. We find this method to be highly precise, but its recall is still considerably lower than those of the LLMs.

- **OpenStreetMap (OSM) + Nominatim API**: A major concern with the NER tagging algorithm is selecting the right geodatabase (or Gazetteer), which links the string tagged to its exact location and country. In many instances, captions reference small towns, roads and buildings that commonly used Gazetteers do not cover and hence we miss out on tagging those captions. To access a larger geographic knowledge base, particularly OpenStreetMap (OSM [9]), we incorporate and test the Nominatim API [10] in our geo-profiling pipeline, which uses OSM to detect locations on Earth. We include this step after the Spacy NER tagging, filtered on both 'GPE' and 'LOC' (Non-GPE locations) tags. This Spacy NER with Nominatim pipeline does provide improved results but posed difficulty in scaling due to restrictions on bulk usage of nominatim.

- **Geoparsepy**: We additionally employ the Geoparsepy (Middleton et al., 2018) to extract country names from captions. As explained in Section 2, it is a geoparser that utilizes the OSM (OpenStreetMap) database to predict the geospatial characteristics of a given text. When applied on our annotated dataset, we find it to successfully extract location mentions in captions in multiple cases, but fail to map those locations to their countries. Hence, we query the Geonames (GeoNames database) database on the location mentions for which the system fails to output a country name. Overall, we find that the method is inferior, compared to the LLM-based methods.

- **Geograpy3**: Geograpy3 [11] is a Python library that extracts place names from text. It internally uses NLTK [12], Wikidata [13] and other resources to recognise entities and disambiguate regions based on population. Although it offers user-friendly APIs for easy access to such information, its evaluation on our 1000 caption dataset shows inadequate results.

The performance of each of these methods is shown in Table 9.

**Generalization of the Mixtral LM to datasets with marginalized countries**. It is to be understood that the 1000 caption dataset that we annotate to evaluate the performance of the Mixtral model is

---

[9] https://www.openstreetmap.org/
[10] https://nominatim.org/
[11] https://pypi.org/project/geograpy3/
[12] https://www.nltk.org//
[13] https://www.wikidata.org/wiki/Wikidata:Main_Page

| Method | Without No Country Class | |
|---|---|---|
| | Precision | Recall |
| Mixtral | 0.86 | 0.82 |
| LLama3.1-8B-Instruct | 0.85 | 0.71 |
| Gemma-2-9b-it | **1.00** | 0.04 |
| GPT4-o | 0.87 | **0.91** |
| NER Tagger | 0.71 | 0.59 |
| NER Tagger + Mixtral | 0.91 | 0.61 |
| String Matching | 0.12 | 0.79 |
| Spacy NER with Nominatim API (OSM) | 0.83 | 0.64 |
| Geoparsepy | 0.67 | 0.26 |
| Geograpy3 | 0.69 | 0.21 |

Table 9: **Performance of different geo-profiling methods.** We find that the Mixtral model outperforms all other methods (except gpt4-o) by a large margin, both in terms of precision and recall. As it is open-source and effective, we choose this model to geo-profile the captions.

randomly sampled from the LAION2B-en captions, and likewise the majority of its captions are from countries like the UK, the US, Canada, Australia etc. We also demonstrate the effectiveness of the Mixtral LM in identifying marginalized countries by extracting 1660 sentences about 10 countries (Tuvalu, Kiribati, Algeria, Yemen, Sao Tome and Principe, Botswana, Tajikistan, Uruguay, Guyana and Malta) and their cities from Wikipedia, and evaluating the model on this dataset. To construct such a dataset, we first identify cities within each marginalised country with a population exceeding 5,000 using the GeoNames database (GeoNames database), ensuring comprehensive coverage across diverse marginalised locations. We then extract Wikipedia pages dedicated to these cities, leveraging the extensive repository accessible via Wikipedia APIs. From the retrieved pages, we systematically extract sentences containing explicit mentions of the respective city names. The resulting dataset offers a robust foundation for testing and analysing geoprofiling capabilities of the GeoProfiler. We find that the Mixtral model effectively identifies the country correctly for 76% of the captions, which shows the efficacy of the chosen model in identifying a wide range of nations from captions.

**Examples of the different Captions**. We show examples of captions with and without location mentions in Table 10. It shows how some captions are underspecified, and while others may contain place mentions, it may not be trivial to extract them and find the associated country names, as evident in Table 9.

## A.5 CONTINENT-WISE ANALYSIS OF GEOGRAPHICAL DISTRIBUTION

In this section, we describe in detail the distribution of the continents across each entity. For 6 of the 10 entities, we find North America (NA) to have the highest frequency, followed by Europe (EU). On the other hand, Europe dominates the distribution for the remaining 4 entities. Asia (AS) is the third most represented continent, followed by Oceania (OC), Africa (Af) and South America (SA). The detailed frequencies per entity and per continent are shown in Table 11.

## A.6 FURTHER DETAILS ON GEOPROFILING IMAGES

In this subsection, we explore the feasibility of geo-profiling the images directly. Firstly, we create a small dataset for each entity, having equal number of images (100) from each country. We evaluate two state-of-the-art image geolocalization methods using these datasets for every entity - a) GeoCLIP (Vivanco et al., 2023), b) OSV-5M (Astruc et al., 2024). While both these methods return GPS coordinates for any given image, we convert them to countries and continents with the help of the Nominatim API. The results are presented in Table 12. We find that GeoCLIP performs better overall than OSV-5M, but its country accuracy is still low. Also, the continent accuracies of GeoCLIP suffer for indoor entities like toilet and kitchen. Overall, we feel that geoprofiling images is highly challenging, hence, we infer countries based on captions as mentioned in the main paper.

|  | Specified | Unspecified |
|---|---|---|
| House | "Thumbnail 4 bed detached house for sale in Southfields, Rochester" | "Exterior house colors with brown roof 04" |
| Flag | "Wooden Framed HOME American Flag" | "Medieval knight on horse carrying a flag - Vector..." |
| Car | "Car on Rent in Vadodara with Driver" | "Under a Car Stock Photography" |
| Kitchen | "Kitchen Countertop Suppliers Calgary" | "Kitchen and dining space" |
| Road | "How Calabar-Odukpani Road Dualization Caused Five Accidents Within A Week" | "nature, road, and forest image" |
| Beach | "Happy boys at Copacabana Beach" | "Abstract background of sand at the beach" |
| Hotel | "Jade Court Motor Lodge, hotel in Hokitika" | "Reception of a hotel with a bell, 3d illustration" |
| Toilet | "Rules for using the toilet in Sochi" | "A toilet used as exhibition space" |
| Bedroom | "Four Bedroom House In Ntinda For Rent" | "Master bedroom with King Bed" |
| Apartment | "Rental apartment Toulouse 758€ CC - Picture 7" | "$3285 Two bedroom Apartment for rent" |

Table 10: **Underspecified and location-specific captions** for each entity

|  | NA | SA | EU | AS | Af | OC |
|---|---|---|---|---|---|---|
| House | 18.76 | 0.53 | **62.37** | 11.14 | 3.21 | 3.99 |
| Flag | **40.71** | 5.70 | 28.84 | 15.54 | 6.54 | 2.67 |
| Car | **38.03** | 0.55 | 34.51 | 19.58 | 3.21 | 4.12 |
| Kitchen | **40.89** | 0.43 | 35.81 | 14.19 | 2.17 | 6.51 |
| Beach | **33.35** | 3.02 | 30.06 | 18.20 | 5.13 | 10.25 |
| Road | 20.72 | 1.55 | **43.01** | 17.59 | 4.33 | 12.79 |
| Hotel | 28.26 | 1.22 | **37.07** | 25.50 | 3.61 | 4.35 |
| Bedroom | 28.79 | 0.82 | **42.78** | 19.86 | 4.00 | 3.76 |
| Toilet | **33.06** | 0.50 | 27.18 | 32.21 | 1.63 | 5.41 |
| Apartment | **54.39** | 0.39 | 24.03 | 16.17 | 2.91 | 2.12 |

Table 11: **Distribution of continents in captions across entities**. We tabulate the representation of the continents across the globe for all 10 entities. As expected, we observe that Europe (EU) and North America (NA) are the most dominant continents, followed by Asia (AS), while Africa (Af), Oceania (OC) and South America (SA) have comparatively lower representations.

## A.7 QUALITATIVE EXAMPLES

We visualize the image-caption pairs after being assigned by the GeoProfiler to their corresponding countries in Figure 7. Specifically, we show images from two high-frequency countries (United States and India), two mid-frequency countries (Brazil and Croatia), and two low-frequency countries (Uganda and Tanzania).

## A.8 ANALYSIS ON MULTILINGUAL IMAGE-CAPTION PAIRS

To analyse the feasibility of extending our approach to multilingual captions, we randomly sample 1000 captions uniformly from 5 languages (i.e., 200 captions for each language) of the LAION2B-multi dataset: French, German, Spanish, Portuguese, and Japanese. For every language, we hire 3 crowdworkers specifically speaking these languages from the Prolific platform (prolific, None) to

| Entity | # of test images | GeoCLIP | | OSV-5M | |
|---|---|---|---|---|---|
| | | Country | Continent | Country | Continent |
| House | 1340 | 0.40 | 0.74 | 0.30 | 0.66 |
| Flag | 3780 | 0.45 | 0.76 | 0.12 | 0.43 |
| Road | 1120 | 0.55 | 0.83 | 0.32 | 0.65 |
| Kitchen | 580 | 0.20 | 0.50 | 0.18 | 0.50 |
| Car | 1140 | 0.32 | 0.66 | 0.16 | 0.51 |
| Beach | 1060 | 0.48 | 0.72 | 0.23 | 0.52 |
| Apartment | 1740 | 0.31 | 0.75 | 0.25 | 0.62 |
| Bedroom | 660 | 0.29 | 0.66 | 0.26 | 0.62 |
| Toilet | 500 | 0.19 | 0.54 | 0.12 | 0.49 |
| Hotel | 1800 | 0.36 | 0.74 | 0.23 | 0.61 |

Table 12: **GeoProfiling Images**. We evaluate the performance of three different models and report the country/continent accuracies for each entity in this table.

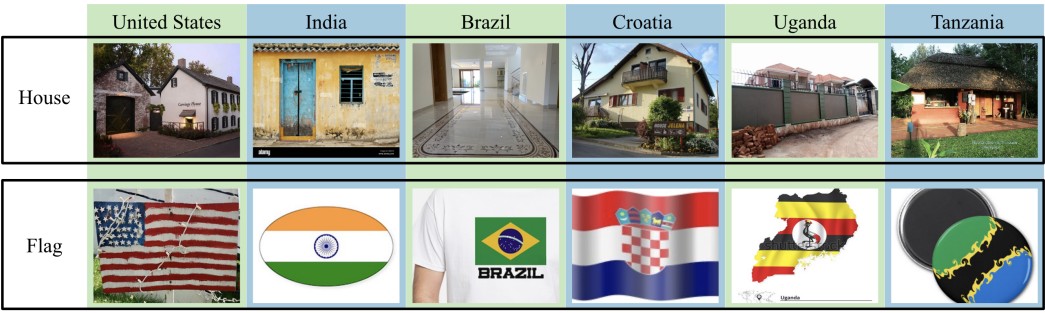

Figure 7: **Images of Houses and Flags** belonging to different countries as predicted by GeoProfiler. For this visual analysis, we pick two high-frequency countries (United States and India), two mid-frequency countries (Brazil, Croatia), and two low-frequency countries (Uganda, Tanzania).

annotate these captions. We thereafter geo-profile these captions using two approaches: (a) applying multilingual LLMs directly (Aya-101 [14] and Bigscience-mt0-xxl[15]), and (b) translating the captions to English with a multilingual LLM (Aya-101) before geo-profiling them using the Mixtral model. The prompt for the Mixtral model is same as mentioned in Appendix A.4. For the multilingual LLMs, we use two kinds of prompts:

- Prompt 1: This is same as that used for the Mixtral model.
- Prompt 2: "You are a geotagging agent who tags each given text to a country.  Given the text 'y',The only output you give is either the coutry name or 'NO' in case the text cannot be tagged to a country"

We observe that while the mixtral outputs of the English-translated captions have a high precision and recall for 3 out of the 5 languages, the performance drops for German and Japanese. Further, this algorithm depends on the quality of translation to English, and may not generalize to low-resource languages. On the other hand, the precision and recall scores of the multilingual models are highly inconsistent, which shows that more research is required in the domain of multilingual data. We present these results in Table 13.

**GeoProfiling English-Translated Multilingual Captions**. We extend the experiments on geoprofiling translated captions by examining multilingual captions from the LAION2B-multi, a subset of the

---

[14]https://huggingface.co/CohereForAI/aya-101
[15]https://huggingface.co/bigscience/mt0-xxl

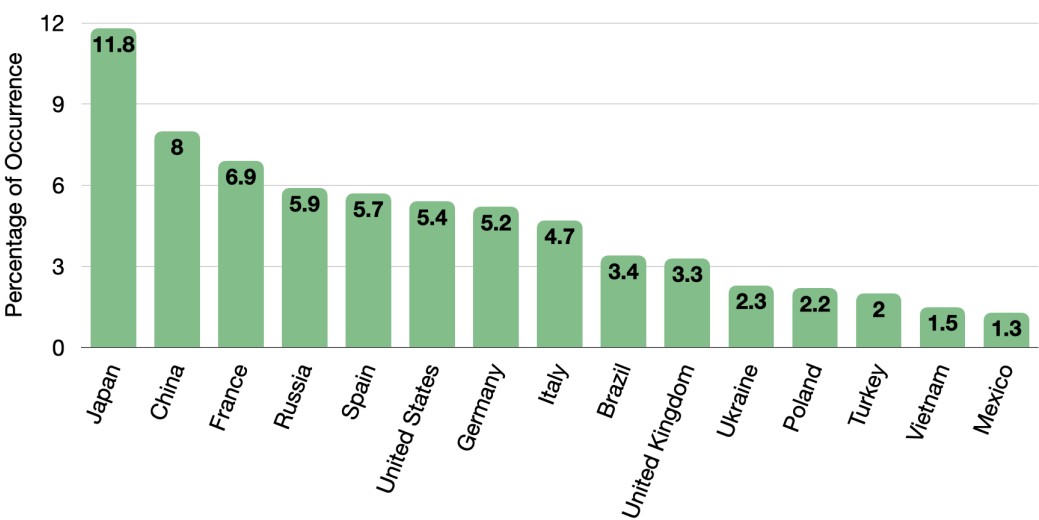

Figure 8: **Percentage of Occurrence of the Top** 10 **countries represented by Multilingual Captions**
The given distribution is based on small dataset of English-translated 10, 000 randomly sampled
captions from the multilingual subset of the LAION dataset.

| Language | Metric | Mixtral | Bigscience-1 | Bigscience-2 | Aya-1 | Aya-2 |
|----------|--------|---------|--------------|--------------|-------|-------|
| German | P | 0.68 | 0.90 | 0.73 | 0.53 | 0.50 |
| | R | 0.91 | 0.21 | 0.35 | 0.79 | 0.12 |
| French | P | 0.82 | 0.92 | 0.89 | 0.61 | 0.96 |
| | R | 0.85 | 0.43 | 0.39 | 0.86 | 0.22 |
| Spanish | P | 0.80 | 0.84 | 0.93 | 0.66 | 0.88 |
| | R | 0.95 | 0.46 | 0.47 | 0.85 | 0.19 |
| Portuguese | P | 0.79 | 0.76 | 0.87 | 0.62 | 0.79 |
| | R | 0.89 | 0.38 | 0.31 | 0.83 | 0.21 |
| Japanese | P | 0.51 | 0.72 | 0.50 | 0.42 | 0.50 |
| | R | 0.87 | 0.55 | 0.18 | 0.88 | 0.02 |

Table 13: **Performance of the Mixtral model and the multilingual LLMs on the multilingual
data**. We find that the Mixtral model outperforms all other methods for most languages, though
the performance drops for German and Japanese. Bigscience-1 and Aya-1 refer to the results of the
Bigscience and Aya LLMs with prompt 1, similarly Bigscience-2 and Aya-2 refer to the results of
those with prompt 2. The multilingual LLMs show uneven performance, rendering them unsuitable
for geo-profiling multilingual captions. (P: Precision, R: Recall)

LAION-5B (Schuhmann et al., 2022) containing 100 languages, with the top 10 (Russian, French,
German, Spanish, Chinese, Japanese, Italian, Portuguese, Dutch, Polish) comprising 56% of the data.
African and South American data remain underrepresented (similar to our observations in the paper).
From a random sample of 10, 000 captions translated into English via the Aya-101 model, 70% were
tagged as underspecified, with Japan and China dominating the location-specific tags. The top 15
countries (Fig. 8) correspond closely to the dataset's most frequent languages. Notably, 73.9% of
location-specific captions are from Asia and Europe, while only 7.4% are from Africa and South
America. These findings demonstrate that our framework is well-suited for geoprofiling multilingual
datasets for high resource languages, making it an important extension of this work.

| r | Represen-tation | Hou-se | Flag | Car | Kit-chen | Bea-ch | Road | Ho-tel | Bed-room | Toi-let | Apart-ment |
|---|---|---|---|---|---|---|---|---|---|---|---|
| 2 | Under | **62.6** | **77.2** | 39.4 | **63.6** | 34.0 | 35.8 | 50.6 | **52.1** | **63.3** | **62.1** |
|   | Over | 12.9 | 4.2 | 19.4 | 15.4 | 34.0 | 26.4 | 18.2 | 26.5 | 18.3 | 11.4 |
| 3 | Under | **56.5** | **68.8** | 26.8 | **58.7** | 38.7 | 23.4 | 41.8 | 46.6 | **55.5** | **55.0** |
|   | Over | 8.0 | 2.1 | 13.4 | 11.9 | 28.0 | 19.0 | 14.4 | 22.4 | 13.9 | 8.4 |
| 5 | Under | 41.7 | **53.4** | 20.6 | **51.8** | 30.0 | 8.8 | 32.5 | 42.3 | 44.0 | 43.9 |
|   | Over | 5.8 | 1.1 | 7.9 | 9.1 | 22.0 | 14.5 | 5.8 | 12.8 | 7.3 | 5.3 |

Table 14: **Geographical Representation in each entity for different values of r**. We show the percentage of countries that are underrepresented (Under) and overrepresented (Over) for r = 2, r = 3 and r = 5. With increasing r, the percentages of under and overrepresentation increase, as expected. We choose r=3 for the results in the main paper, following previous works (Schwöbel et al., 2023)

## A.9    FURTHER DETAILS ON GEOGRAPHICAL REPRESENTATIVENESS

**Ground Truth Distribution for each Entity**. In the main paper, we motivate the need for comparing the geographical distributions obtained from the dataset with a ground truth reference distribution (subsection 4.2). From each entity, we obtain the ground truth distribution from real-world data. For house (and the related kitchen, bedroom, toilet) we use the available data on number of households in the world (House reference data). For apartment, we use the same data as house, as they are semantically similar. Since flag is a universal concept, we assume its ground truth distribution to be uniform. For car, we use country-wise data on motor vehicles per capita(Car reference data), similarly for road, we use details on road network size (Road reference data). For hotel we use the information provided by UN World Tourism (Hotel reference data). We approximate the number of beaches in a country through its length of coastline available (Beach reference data).

**Additional Findings**. In the main paper, we determine geographical representation of a country by computing the ratio of its probability in the dataset (for a given entity) with its probability in the ground truth distribution. Intuitively, a country can be called overrepresented if this ratio is greater than a threshold r, and similarly it is called underrepresented if this ratio is lower than $\frac{1}{r}$. We show our findings with respect to $r = 3$ in the main paper subsection 4.2. Here we investigate for a few other values of r (specifically r = 2 and r = 5), showing that with higher values of r, the percentages of over and underrepresentation increases (Table 14). The choice of the exact value of r is dependent on the perception of the practitioner who is evaluating the geographical representativeness of a dataset. We select the value to be 3, following related previous works (Schwöbel et al., 2023).

## A.10    FURTHER DETAILS ON DIVERSITY ANALYSIS

We explain the metric we use for measure diversity in subsection 4.3 in the main paper, and share the scores we obtain for the entity images.

In this subsection, we extend our study to the captions, as well as images generated from the Stable Diffusion model respectively.

**Captions:** To compute the score for the captions, we first encode them using the SentenceTransformers embeddings (all-MiniLM-L6-v2) (Sentence Transformers). Similar to the images, we compute diversity scores for captions belonging to a specific country and entity, but only consider countries that have more than 100 image-caption pairs for a given entity. Overall, we find that car and road have the most variations in their captions. The Flag captions are the least diverse since they are mostly descriptions of the entity, whereas captions for other entities often describe surroundings and related context. The detailed scores for each entity are shown in Table 16. We further define $\rho_{fc}$ as the Pearson's correlation coefficient of country-wise frequency values with the diversity scores of these captions, and find that most entities exhibit weak to moderate positive correlation with frequency (see Table 15).

**Generated Images:** We generate 1000 images for each country having frequency $\geq 100$ in the original dataset for each entity, and encode them with DINOv2 (similar to the original images). The average diversity score across all entities is 0.60 for the generated images, as opposed to 0.75 for

| | House | Flag | Car | Kitchen | Beach | Road | Hotel | Bedroom | Toilet | Apartment |
|---|---|---|---|---|---|---|---|---|---|---|
| $\rho_{\text{fc}}$ | -0.21 | 0.42 | 0.24 | 0.39 | 0.47 | 0.36 | 0.40 | 0.38 | 0.03 | 0.07 |
| $\rho_{\text{fg}}$ | -0.31 | -0.37 | 0.04 | -0.48 | 0.2 | 0.34 | 0.08 | -0.32 | -0.47 | -0.34 |

Table 15: **Frequency and diversity correlations for captions and generated images.** For each entity, we compute the correlation between the frequency of countries and diversity scores of a) the captions ($\rho_{fc}$), b) the generated images from Stable Diffusion ($\rho_{fg}$). For generated images, we observe negative correlations for the majority of entities.

| entities | Text | Image | Stable Diffusion |
|---|---|---|---|
| House | $0.76^{\pm 0.05}$ | $0.80^{\pm 0.03}$ | $0.59^{\pm 0.05}$ |
| Flag | $0.65^{\pm 0.04}$ | $0.83^{\pm 0.03}$ | $0.74^{\pm 0.04}$ |
| Car | $0.80^{\pm 0.07}$ | $0.81^{\pm 0.03}$ | $0.64^{\pm 0.04}$ |
| Kitchen | $0.71^{\pm 0.04}$ | $0.61^{\pm 0.03}$ | $0.48^{\pm 0.04}$ |
| Beach | $0.74^{\pm 0.04}$ | $0.73^{\pm 0.03}$ | $0.54^{\pm 0.05}$ |
| Road | $0.79^{\pm 0.04}$ | $0.81^{\pm 0.03}$ | $0.58^{\pm 0.04}$ |
| Hotel | $0.73^{\pm 0.03}$ | $0.80^{\pm 0.02}$ | $0.62^{\pm 0.06}$ |
| Bedroom | $0.71^{\pm 0.04}$ | $0.62^{\pm 0.02}$ | $0.50^{\pm 0.03}$ |
| Toilet | $0.74^{\pm 0.03}$ | $0.67^{\pm 0.04}$ | $0.62^{\pm 0.06}$ |
| Apartment | $0.68^{\pm 0.07}$ | $0.78^{\pm 0.03}$ | $0.69^{\pm 0.04}$ |

Table 16: **Entity-wise diversity scores** for the training captions and images, and the generated images from Stable Diffusion

training images. While our results indicate that generated images have less variation than real images, perhaps this gap can be explained by differences in how captions in the dataset and prompts for Stable Diffusion are written (for an entity $n$ and country $c$, our prompt for Stable Diffusion is "High definition image of a $n$ in $c$"). The detailed results are present in Table 15. Similar to the real images and captions, we define $\rho_{fg}$ as the Pearson's correlation coefficient of country-wise frequency values with the diversity scores of the generated images. We observe that the diversity scores for 6 entities are negatively correlated with frequency. Kitchen has the highest negative correlation ($\rho_{fg} = -0.48$). For example, while $div(\text{Kitchen Images in the US}) = 0.4$ and $div(\text{Kitchen Images in Morocco}) = 0.55$, frequencies of kitchens of US and Morocco are $14177$ and $127$ respectively. Examining further, we find that American kitchens tend to be visually and stylistically more similar to one another than Moroccan kitchens, suggesting that higher frequency countries may have more similar-looking images (both training and generated) than lower frequency ones, leading to negative or no correlations. Detailed results are shown in Table 15.

**Effect of biases in the feature encoders on correlation between frequency and diversity**

We acknowledge that pretrained models used to compute the diversity scores may share biases due to imbalanced training data. For example, models might group images of high-frequency countries more closely than those of low-frequency nations. However, we find that this is generally not true. For instance, countries like the US, UK, China, and India have higher diversity scores than the average across all countries for most entities (see Table 17). Additionally, we evaluate whether diversity scores align with human perception by iteratively showing human participants images from a pair of countries with high and low diversity scores, asking which country's images appear more diverse to them. Based on $150$ responses collected from $5$ participants, we find that human judgments are consistent with diversity scores $82\%$ of the time.

**Qualitative Visualizations**. Further, we show the images of roads from Norway ($div(\text{Norway}) = 0.77$) and Mexico ($div(\text{Mexico}) = 0.83$) as present in the LAION2B-en dataset in Figures 9 and 10. We notice that while the images from Norway mostly show images of roads in various landscapes, the images from Mexico depict other noises like humans and cars, leading to Mexican roads having higher diversity scores than those of Norway. In case of generated images, we see that kitchens of United States, having frequency $14117$, have a diversity score of $0.40$, whereas Morocco, with a frequency of $127$ has a diversity score of $0.54$. Example images can be seen in Figures 11 and 12

| | Apartment | House | Road | Hotel | Car | Flag | Bedroom | Toilet | Beach | Kitchen |
|-------|-----------|-------|------|-------|------|------|---------|--------|-------|---------|
| India | 0.79 | 0.80 | 0.84 | 0.81 | 0.81 | 0.86 | 0.64 | 0.72 | 0.75 | 0.58 |
| China | 0.83 | 0.83 | 0.83 | 0.80 | 0.82 | 0.86 | 0.64 | 0.65 | - | 0.61 |
| US | 0.81 | 0.81 | 0.87 | 0.82 | 0.82 | 0.86 | 0.60 | 0.64 | 0.77 | 0.59 |
| UK | 0.79 | 0.68 | 0.85 | 0.82 | 0.83 | 0.86 | 0.61 | 0.70 | 0.77 | 0.62 |
| Avg | 0.78 | 0.80 | 0.81 | 0.80 | 0.81 | 0.83 | 0.62 | 0.67 | 0.73 | 0.61 |
| Std | 0.03 | 0.03 | 0.03 | 0.02 | 0.04 | 0.03 | 0.02 | 0.04 | 0.03 | 0.03 |

Table 17: **Comparison of diversity scores of high frequency countries** with the average diversity score across all countries, for each entity.

for United States and Morocco respectively. While the images of United States seem visually alike, those of Morocco show more variations. For example, some images are inside kitchens, some are outside. In fact, some images of Morocco do not have kitchens at all, thereby contributing towards a notion of increased diversity at the cost of inaccurate depictions of the entity. Note that the prompt we use for generating the images for any entity $n$ and country $c$ is: "High definition image of a {n} in {c}". Such a simplistic instruction prompts the model to generate the exact entity in question, based on its understanding of the same. Hence, we notice much lesser variations in the generated images than those of the real images.

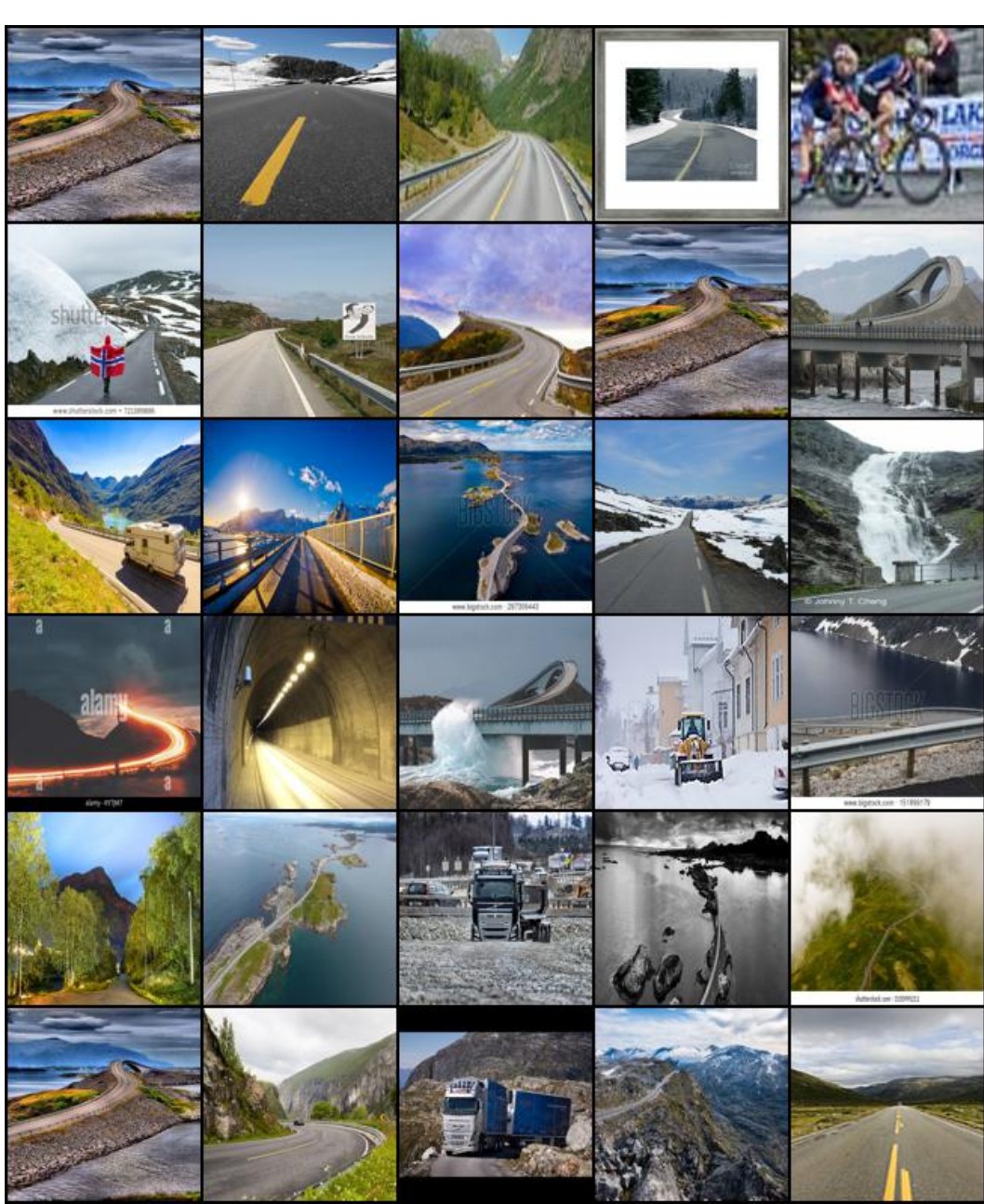

Figure 9: **Training images of Road for Norway.** There are 715 images of roads from Norway. The diversity score for the images is 0.77.

1566
1567
1568
1569
1570
1571
1572
1573
1574
1575
1576
1577
1578
1579
1580
1581
1582
1583
1584
1585
1586
1587
1588
1589
1590
1591
1592
1593
1594
1595
1596
1597
1598
1599
1600
1601
1602
1603
1604
1605
1606
1607
1608
1609
1610
1611
1612
1613
1614
1615
1616
1617
1618
1619

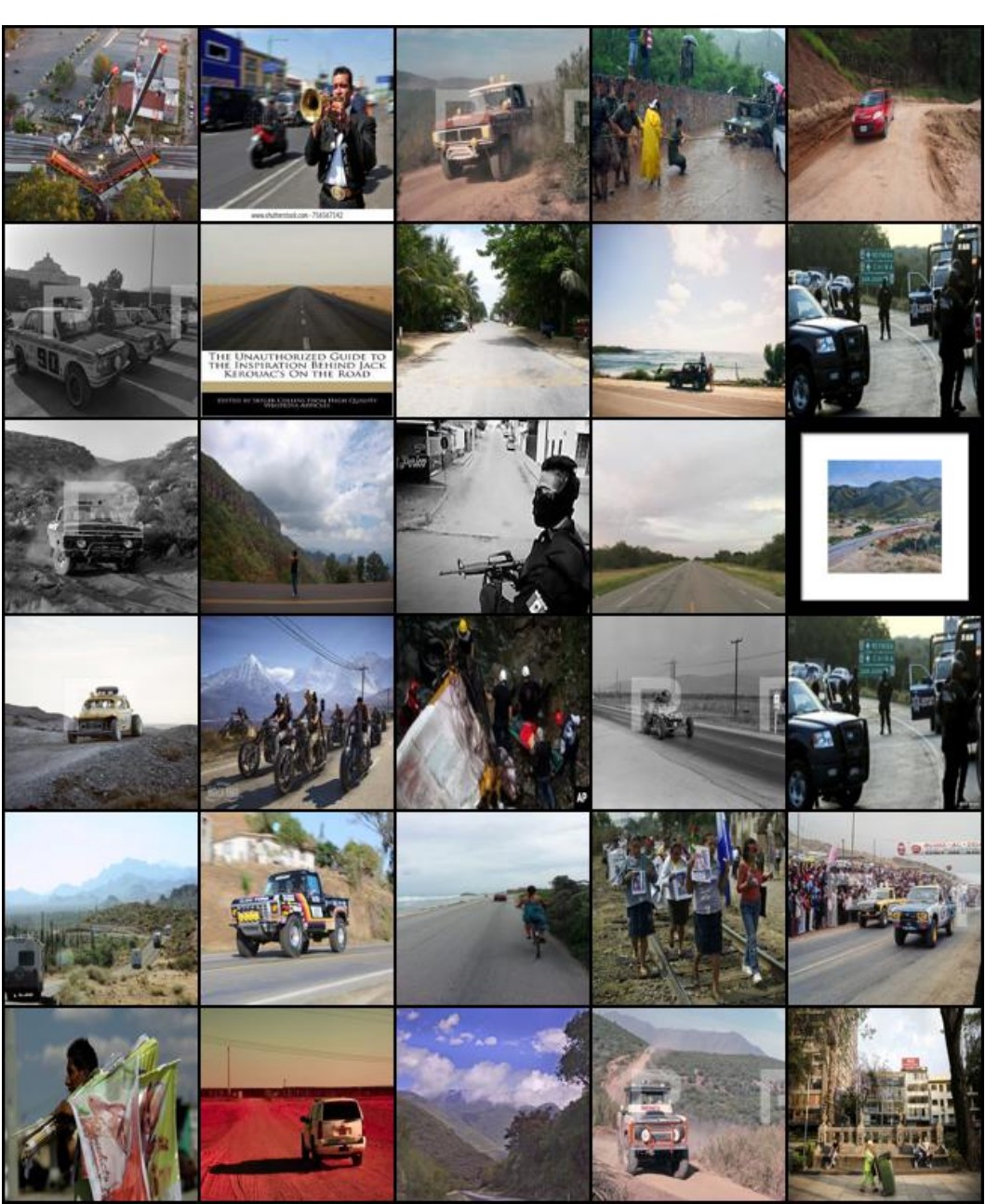

Figure 10: **Training images of Road for Mexico.** There are 294 images of roads from Mexico. The diversity score for the images is 0.83.

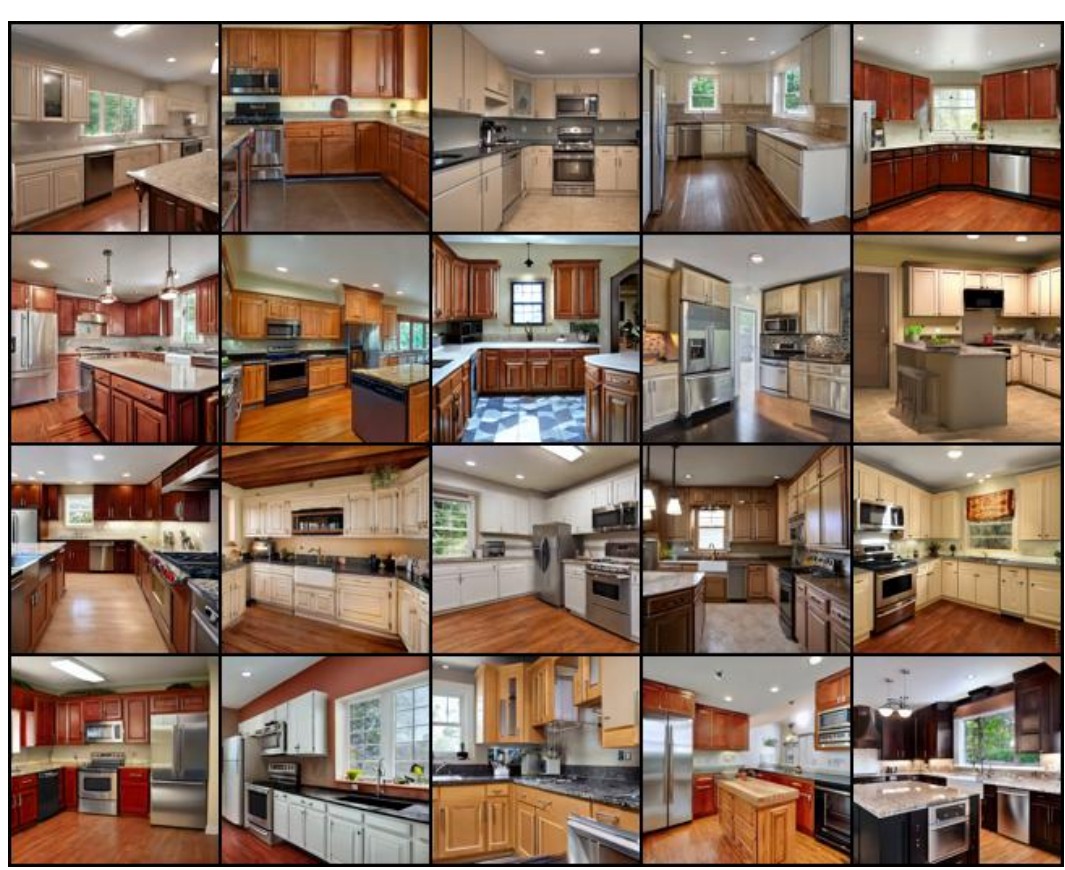

Figure 11: **Kitchen images generated for United States.** While we found 14117 American kitchens in our dataset, the diversity score for generated images from the same country is 0.41.

1674
1675
1676
1677
1678
1679
1680
1681
1682
1683
1684
1685
1686
1687
1688
1689
1690
1691
1692
1693
1694
1695
1696
1697
1698
1699
1700
1701
1702
1703
1704
1705
1706
1707
1708
1709
1710
1711
1712
1713

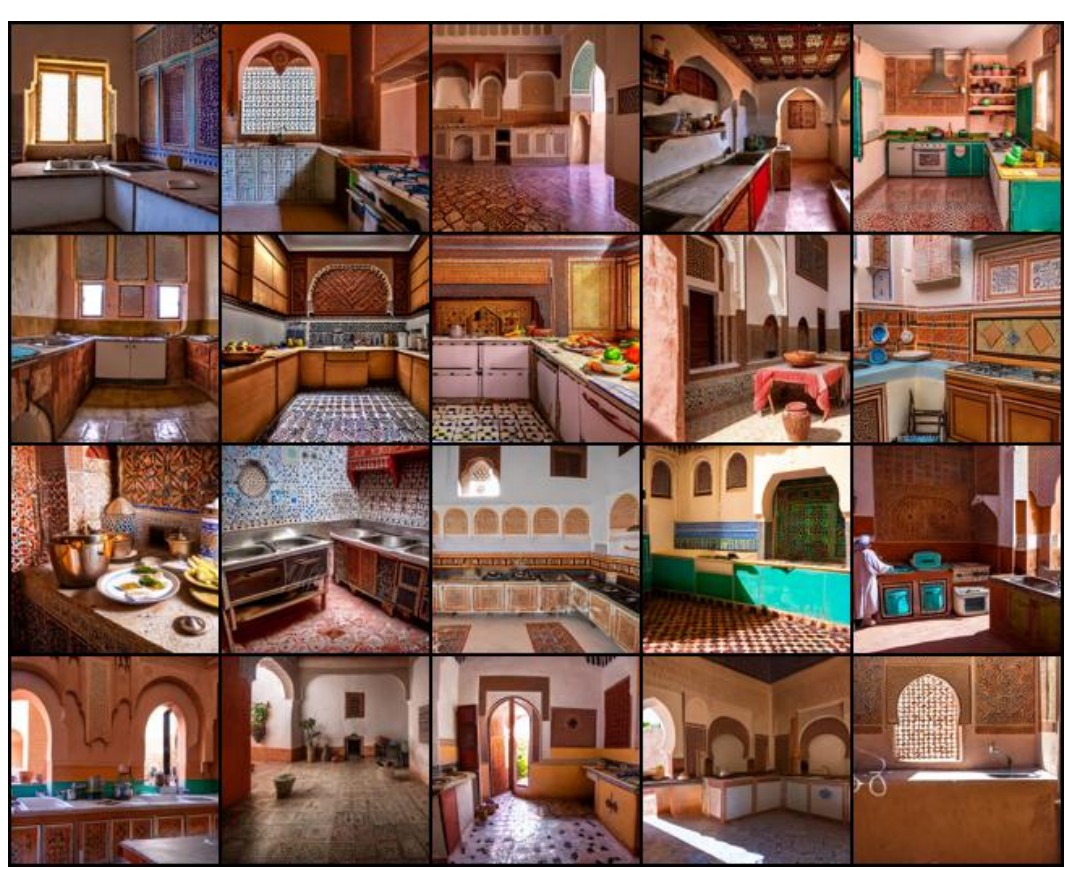

Figure 12: **Kitchen images generated for Morocco.** While we found 127 Moroccan kitchens in our dataset, the diversity score for generated images from the same country is 0.55.
