# OpenReview forum: "Where Do Images Come From? Analyzing Captions to Geographically Profile Datasets"
_ICLR.cc/2025/Conference — Submitted to ICLR 2025_

### Official Review · Reviewer_kdSK · 2024-11-03

**Soundness:** 3
**Presentation:** 4
**Contribution:** 2
**Rating:** 5
**Confidence:** 4

**Summary:**

The paper addresses the issue of geographical biases in vision-language models (VLMs) by investigating the geographical origins of images used in training these models. The authors introduce GeoProfiler, a system designed to map image-caption pairs in multimodal datasets to their corresponding countries based on location information extracted from captions. GeoProfiler employs a large language model (Mixtral-8x7B Instruct) to accurately infer countries from captions, achieving a high precision of 0.86 and recall of 0.82.
The authors apply GeoProfiler to the LAION2B-en dataset, a large-scale dataset with English captions. They focus on 10 globally relevant entities (e.g., house, flag, car) to analyze the geographical distribution of the data. The key findings echo previous works in this area and include:
- The geographical distribution of images is highly skewed, following a power law distribution in 8/10 entities. The United States, the United Kingdom, and India are the most represented countries, accounting for 53.7% of the samples.
- African and South American countries are significantly underrepresented, constituting only 2.0% and 4.3% of the images, respectively
- There is a strong positive correlation (rho = 0.79) between a country's GDP and its frequency in the dataset, indicating that wealthier countries are more represented.
- An analysis of the diversity of images from individual countries reveals that a higher number of images does not necessarily imply greater diversity.
- A 46.1% of captions lack geographical information.

The paper highlights the limitations of current methods in determining image sources and emphasizes the importance of creating geographically representative datasets. GeoProfiler is a tool aimed to help data curators and practitioners measure and improve the geographical diversity of datasets.

**Strengths:**

- The research is methodologically sound and exhibits a good level of rigor. The authors systematically develop GeoProfiler, starting with an analysis of simple baselines like string-matching and Named Entity Recognition (NER), and demonstrating their limitations in this context. By employing the Mixtral-8x7B Instruct model, they achieve a high precision of 0.86 in mapping captions to countries. The application of GeoProfiler to the LAION2B-en dataset is thorough, involving the analysis of 1 million image-caption pairs for each of the 10 selected entities. The statistical analyses, including the correlation with GDP (rho = 0.79) and the power law distribution fitting lead to some insightful findings. The appendix provides extensive methodological details, particularly about the entity-presence filtering process, the creation of the annotation dataset, and the inter-annotator agreements (Appendix A.3).
- The paper is well-written and clearly structured. Each component of GeoProfiler is explained in detail, and the methodology is presented in detail, facilitating reproducibility. The use of figures and tables, such as the world maps showing the distribution of countries for specific entities and the correlation graphs with GDP, illustrate the key points. Additionally, the authors provide qualitative examples and comprehensive appendices that enhance the reader's understanding of their approach and findings.
- This work holds significance in the domains of AI bias, data ethics, and responsible AI development. By studying and quantifying the geographical biases present in large-scale datasets like LAION2B-en, the paper provides insights into how such biases can propagate into VLMs and affect their deployment globally. The strong correlation between country representation and GDP indicates socio-economic disparities reflected in AI training data. These findings emphasize the need for more geographically diverse datasets to ensure that AI models are representative of different regions around the world. GeoProfiler could potentially serve as a practical tool for data curators, practitioners, and auditors in their efforts to assess and improve geographical diversity in datasets.

**Weaknesses:**

- Unfortunately, the authors seem to disregard a large body of pre-LLM work on geotagging multimodal data [1,2,3,4]. Their baseline approaches are quite simplistic (even though generally quite effective), while there are numerous text-based approaches that perform very accurately even at the level of city prediction. As a result, I have strong doubts on the accuracy and efficiency of GeoProfiler compared to pre-LLM state-of-the-art methods [2, 4].

- While the authors have tried to extend their analysis to multilingual captions, the evaluation remains preliminary. The challenges faced suggest that the method may not generalize well to all languages.

- The authors limit their analyses to only 10 entities, which might itself constitute yet another source of bias. It would be interesting to include at least one entity containing humans, which might also offer fruitful ground for more in-depth analysis of representation bias.

- The authors have made some efforts to include and analyze data from underrepresented regions, however, details about these regions remain limited.

- The authors do not delve deeper into socio-economic factors beyond the correlation with GDP.

- No consideration is made of when the images were created/uploaded. The temporal aspect in the analysis could offer additional insights.

[1] Middleton, et al. (2018). Location extraction from social media: Geoparsing, location disambiguation, and geotagging. ACM Transactions on Information Systems (TOIS), 36(4), 1-27.

[2] Kordopatis-Zilos, et al. (2017). Geotagging text content with language models and feature mining. Proceedings of the IEEE, 105(10), 1971-1986.

[3] Luo, et al. (2011). Geotagging in multimedia and computer vision—a survey. Multimedia Tools and Applications, 51, 187-211.

[4] Hu, et al. (2023). Location reference recognition from texts: A survey and comparison. ACM Computing Surveys, 56(5), 1-37.

**Questions:**

- How did the authors decide to select the ten specific entities for their analysis, and do they believe this selection fully captures the diversity in the dataset? Could the limited number of entities impact the generalizability of the findings?
- How do the authors address the underrepresentation of marginalized or less-documented regions in your analysis? Could this underrepresentation affect the generalizability of the findings?
- Consider including pre-LLM state-of-the-art methods in the experimental analysis.
- Consider expanding the multilingual analysis further by addressing the challenges observed with certain languages and exploring more diverse language models or translation techniques.
- Providing examples or case studies using accessible datasets would enhance the impact and usability of your tool.

**Details Of Ethics Concerns:**

The paper processes a large-scale dataset (LAION) that is known to suffer from a number of harmful issues. While the authors argue that their sample is small and the probability of including harmful images there does not raise issues, I would expect a more in-depth discussion. Additionally, potential errors in the performance of the GeoProfiler may lead to wrong conclusions regarding the extent/degree of biases in certain datasets.

---

### Official Review · Reviewer_uwaN · 2024-11-04

**Soundness:** 3
**Presentation:** 3
**Contribution:** 2
**Rating:** 6
**Confidence:** 3

**Summary:**

This study proposed GeoProfiler which geographically profiles multimodal datasets by mapping image-caption pairs to countries. GeoProfiler was then applied to geographically profile the English captions of the LAION dataset for 10 common entities. The dataset was not considered diversified because some countries were severely under-represented. A high correlation between a country's GDP and frequency was observed.

**Strengths:**

(1) This study proposed a unique perspective for understanding the biases in vision-language models.

(2) Multiple experiments were conducted to analyze the biases in geographic distributions associated with the image-caption pairs.

**Weaknesses:**

(1) The contributions of the study were not well articulated. As the authors discussed in the related work section, there have been many efforts in geographically profiling datasets. Some of them focus on visual datasets, and some of them focus on textual datasets. However, the authors claimed that the uniqueness of this study lies in the focus on vision-language datasets. It is unclear why the proposed study is different from the existing methods for profiling datasets such as MS-COCO.

(2) The analysis was only conducted for one dataset. The findings of the study may not be general.

(3) GeoProfiler is composed of many modules. The motivation for each module is clear. However, the study could be more robust and the contributions could be clearer if for each process, the authors could compare their design with baselines.

**Questions:**

Questions:

(1) Are there cases where the country can only be inferred based on the image instead of the caption? How would GeoFilter handle that case?

(2) Based on the current findings, what would be the suggestions for addressing the potential biases in the vision-language models?

Minor:

(1) The font size of the text in Table 1 can be reduced so that no hyphen will be needed.

---

### Official Review · Reviewer_8PS9 · 2024-11-04

**Soundness:** 3
**Presentation:** 3
**Contribution:** 3
**Rating:** 6
**Confidence:** 3

**Summary:**

This paper proposes a system called "GeoProfiler," which maps image-caption pairs in vision-language datasets to corresponding countries. Specifically, the study analyzes the geographical distribution of 10 entities using the LAION2B-en dataset, which contains images with English captions. The results show that 46.1% of all captions are underspecified, and the geographical distribution of eight entities follows a power law distribution. Additionally, it was found that countries such as the United States, the United Kingdom, and India are frequently represented, while countries in South America, Africa, and Oceania are underrepresented. Furthermore, an investigation into the country-wise diversity of images for each entity revealed that frequency does not correlate with diversity.

**Strengths:**

- The proposed method named GeoProfiler leverages the latest large language models to extract geographical information, enabling context-aware profiling. As a tool proposed for analytical purposes, it holds considerable value.
- The study presented in this paper is significant in that it quantifies the geographical biases present in datasets, aiding in the creation of more fair and balanced datasets.
- The analysis is backed by statistical validation, which lends a certain degree of reliability to the presented results.

**Weaknesses:**

- The investigation is limited to a dataset with English captions, which may result in a biased assessment of geographical representation.
- While GeoProfiler is an interesting tool, it remains primarily a methodological approach that combines existing technologies.
- The potential biases inherent in the LLMs and image classification models used are not deeply explored, and the discussion on these limitations is insufficient.

**Questions:**

- Due to limitations in analysis accuracy, this study focuses only on English data. However, how do you anticipate the differences in conclusions between a study limited to English and one that includes multiple languages? Also, where do you see the value in a study that targets only English?
- It is possigle that biases inherent in the LLMs and image classifiers used in GeoProfiler could influence the results of geographical profiling. What are your thoughts on this issue?
- This analysis is limited to the LAION2B-EN dataset. Why did you choose not to include multiple datasets in your analysis? Additionally, how do you think the conclusions might differ if a multi-dataset approach were adopted?

---

### Meta-Review · Area_Chair_Mvxy · 2024-12-19

**Metareview:**

This paper introduces "GeoProfiler”,  a system for mapping image-caption in multimodal visio-linguistic datasets to corresponding countries based on information extracted from captions. GeoProfiler utilises a LLM (Mixtral-8x7B Instruct) to infer origin countries from captions with high-accuracy. The authors use GeoProfiler to geographically map the LAION2B-en dataset for 10 globally relevant entities they identify. The findings reveal overrepresentation of the US, UK and India accounting for 53% of the LAION2B-en image-text pairs while African and South American countries are underrepresented, accounting only for 2.0% and 4.3% images respectively.

An important contribution with interesting insights into the LAION2B-en dataset’s corresponding geographical origins.

Well written and clearly structured paper with sound analysis and interesting observations into the dataset.

The paper is unfortunately limited in scope as the broader implication of geographical profiling is not considered beyond stating it encourages transparency. For example, what are the benefits and potential risks for a given country that might emerge from under or over representation of that country in training data?

**Additional Comments On Reviewer Discussion:**

Issues around generalizability of findings and analysis on single dataset: the authors address these sufficiently by emphasizing that their findings are dataset dependent and by adding further analysis of a small portion from the Datacomp dataset.

Biases inherent in LLMs and image classifiers used in GeoProfiler potentially influencing the results of geographical profiling: this is a valid concern that has not been satisfactorily addressed by the authors.

Lack of novelty: satisfactory explanation outlining how the current contribution differs from previous work and methods used for profiling other datasets such as MS-COCO.

Inferring country based on image instead of caption: the authors show that this is highly ineffective. Overall informative discussion.

Better articulation of the motivation for each module of the proposed method: addressed satisfactorily.

Missing relevant previous work: addressed satisfactorily.

---

### Decision · Program_Chairs · 2025-01-22

Reject